# The global flood protection savings provided by coral reefs

Michael W. Beck [1,2], Iñigo J. Losada[3], Pelayo Menéndez[3], Borja G. Reguero[1,2], Pedro Díaz-Simal[3] & Felipe Fernández[3]

Coral reefs can provide significant coastal protection benefits to people and property. Here we show that the annual expected damages from flooding would double, and costs from frequent storms would triple without reefs. For 100-year storm events, flood damages would increase by 91% to $US 272 billion without reefs. The countries with the most to gain from reef management are Indonesia, Philippines, Malaysia, Mexico, and Cuba; annual expected flood savings exceed $400 M for each of these nations. Sea-level rise will increase flood risk, but substantial impacts could happen from reef loss alone without better near-term management. We provide a global, process-based valuation of an ecosystem service across an entire marine biome at (sub)national levels. These spatially explicit benefits inform critical risk and environmental management decisions, and the expected benefits can be directly considered by governments (e.g., national accounts, recovery plans) and businesses (e.g., insurance).

[1] The Nature Conservancy, University of California, 115 McAllister Way, Santa Cruz, CA 95060, USA. [2] Department of Ocean Sciences, University of California, Santa Cruz, CA 95060, USA. [3] Environmental Hydraulics Institute "IHCantabria", Universidad de Cantabria, Santander, Cantabria 39011, Spain. Correspondence and requests for materials should be addressed to M.W.B. (email: mbeck@tnc.org)

The impacts of coastal flooding are substantial and growing given population growth, coastal development and climate change[1–4]. Unfortunately, these risks are often discounted in development choices[5,6]. Coastal development also causes losses in coastal habitats, which will further heighten risks[7–10]. There is a pressing need to advance risk reduction and adaptation strategies to reduce flooding impacts[4,6,11].

Coral reefs serve as natural, low-crested, submerged breakwaters, which provide flood reduction benefits through wave breaking and wave energy attenuation. These processes are functions of reef depth and secondarily rugosity[12–16]. The flood reduction benefits of coral reefs and other coastal habitats are predicted to be high and even cost effective in comparison to traditional approaches[13,17–19].

Reefs have experienced significant losses globally in living corals and reef structures from coastal development; sand and coral mining; overfishing and destructive (e.g., dynamite) fishing; storms; and climate-related bleaching events[8,20–23]. There is clear

evidence of reef flattening globally from the loss of corals and from the bioerosion and dissolution of the underlying reef carbonate structures[14,24–27]. Not all reefs are declining, and reefs can recover from bleaching, overfishing and storm impacts, but the overall pattern of significant losses across geographies is clear[20,21]. Scientists and international agencies, including the Intergovernmental Panel on Climate Change and the World Bank, have expressed grave concern about the current and future condition of coral reefs, and the loss of the benefits they provide[28–30].

Although reefs and other coastal habitats can provide flood protection benefits, they are rarely accounted for directly in coastal management, because these services are not quantified in terms familiar to decision-makers, such as (loss of) annual expected benefits[12]. Provisioning services such as fish or timber production, which represent products that are harvested from ecosystems, have been valued globally and considered in resource management decisions[31,32]. Regulating services, which represent benefits provided when ecosystems are left intact such as flood

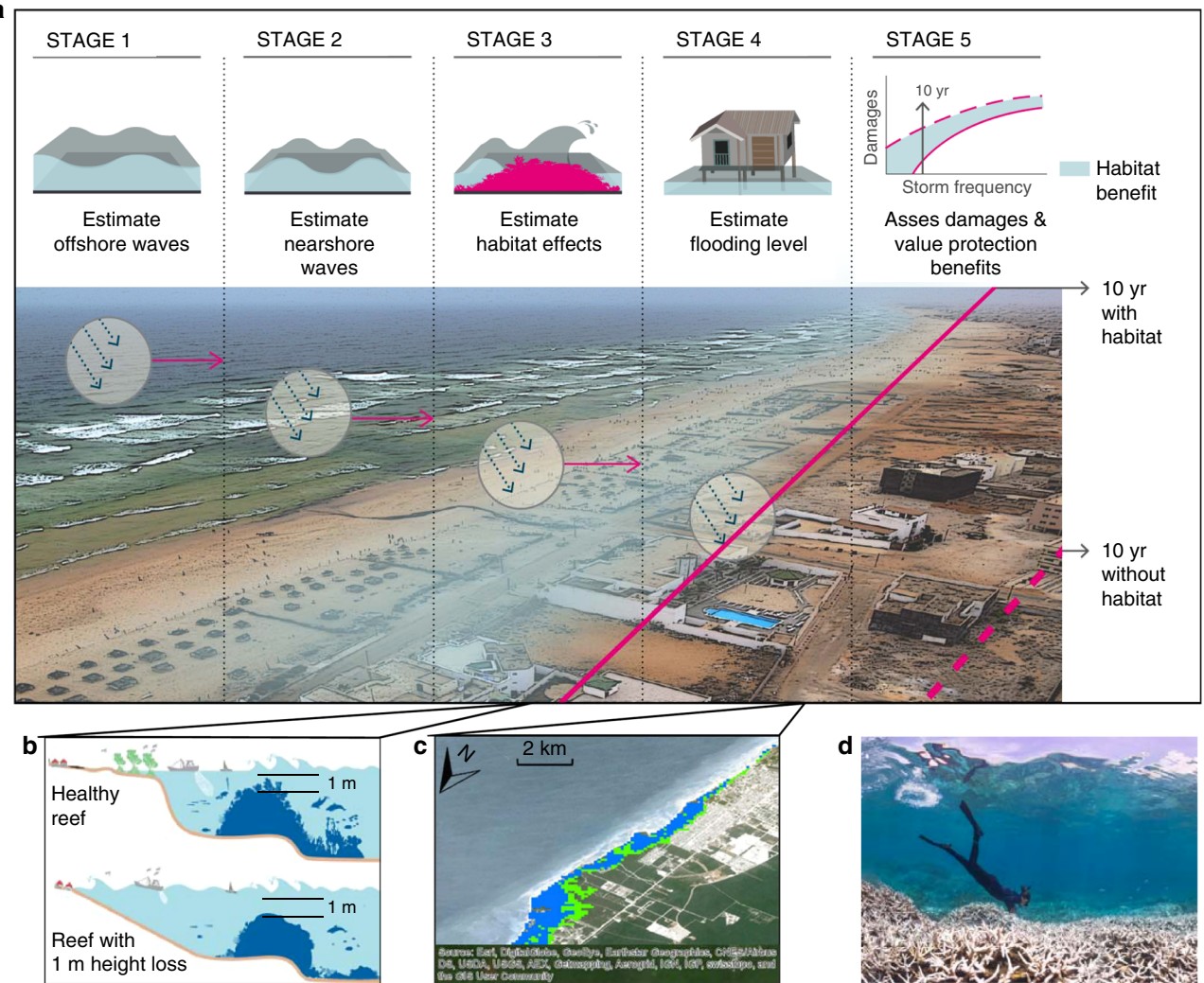

**Fig. 1** Key steps and data for estimating the flood protection benefits provided by reefs. **a** Stage 1: Oceanographic data are combined to assess offshore sea states (waves and sea level). Stage 2: Waves are modified by nearshore hydrodynamics. Stage 3: Effects of habitat on wave run-up are estimated. Stage 4: Flood heights are extended inland along profiles (every 2 km) for four locally generated, storm events (10, 25, 50, 100-yr events) with and without coral reefs. Stage 5: The land, people and built capital damaged under the flooded areas are estimated. Image © TNC. **b** The scenarios for reef loss only assume a loss of the top 1-m in height and roughness across the reef profile. **c** Example results for Mayan Riviera in Mexico; blue polygons are expected flooding in 25-yr event and green polygons are added flooding without the top 1 m of reefs. Map Data © 2018 Google. **d** Inset photo shows coral reef bleaching of top most branching corals in 2015 El Nino event in Guam. ©The Ocean Agency/XL Catlin Seaview Survey

and erosion reduction, have rarely been rigorously valued globally using process-based models, although there is work towards this end[33,34].

Better valuations of the protection services from coastal habitats could inform decisions to meet multiple objectives in risk reduction and environmental management[35–38]. One important pathway through which these services may be considered is in national economic accounts[35]. The United Nations has identified a general approach for assessing ecosystem services in these accounts[39]. The World Bank has developed guidelines for how these approaches could be applied to assess risk reduction benefits of coral reefs to inform decisions on coastal zone management, development loans, and adaptation grants[12].

Natural flood protection benefits are amenable to spatially explicit quantification, because of the broader work on assessments of flood risks and artificial coastal defenses. Robust, process-based flooding models are widely used in the engineering and insurance sectors to inform risk management and development decisions. These process-based models value benefits by comparing the flood damages avoided in scenarios with and without structures (e.g., seawalls or reefs)[12,40]. These models have recently been used to quantify ecosystem benefits in local studies (e.g., within bays) and in a couple of national studies[41–44]. However these flooding studies do not provide a probabilistic assessment of economic risk at any scale.

Using process-based flooding models, we estimate the annual expected benefit of coral reefs for protecting people and property globally. Building on earlier methods and recommended approaches[3,12,45], we compare flooding for scenarios with and without reefs for four storm return periods. The without reefs scenarios assume only a decrease of 1 m in the height and roughness of coral reefs. We estimate the land, population and built capital flooded across all coastlines with coral reefs to a 90 m resolution (Fig. 1). We then derive the annual expected benefit of coral reefs for flood damage reduction from local to global levels.

## Results

**Reefs and global flood reduction benefits.** Globally, reefs avert substantial flood damages and thus provide significant annual expected benefits for flood protection. Across reef coastlines (71,000 km), reefs reduce the annual expected damages from storms by more than $4 billion. Without reefs, annual damages would more than double (118%) and the flooding of land would increase by 69% affecting 81% more people annually (Fig. 2). Reefs provide more benefit for lower intensity, frequent storms, but even during more extreme events the benefits of reefs to people and property are substantial (Fig. 3, Supplementary Fig. 1). For 25-year events, reefs reduce flooding for more than 8700 km² of land and 1.7 million people, and provide $36 billion in avoided damages to built capital (Fig. 3, Supplementary Fig. 2). For 100-year events, the topmost 1 m of reefs provide flood reduction benefits that result in $130 billion in avoided damages (Fig. 3). Without reefs, damages would increase by 90% for 100-year events and 141% for 25-year events.

**Effects of climate change.** Future sea level rise will increase risks, and these risks will be even greater if reefs are lost too (Fig. 4). For example in 2100, the land flooded under a 100-year storm event increases by 64% under a business-as-usual (high) emissions scenario (RCP 8.5) with no reef loss. If this relative sea level rise is coupled with a 1 m loss in reefs, the land flooded increases by 116% (Fig. 4).

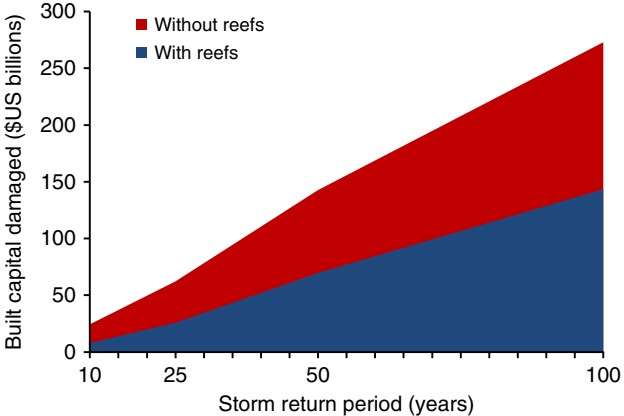

**Fig. 3** The expected economic benefits of coral reefs for flood protection in avoided damages. The values are the expected damages to global built capital from flooding with and without reefs by storm return period. The difference between the curves represents the avoided damages or benefits provided by reefs at present

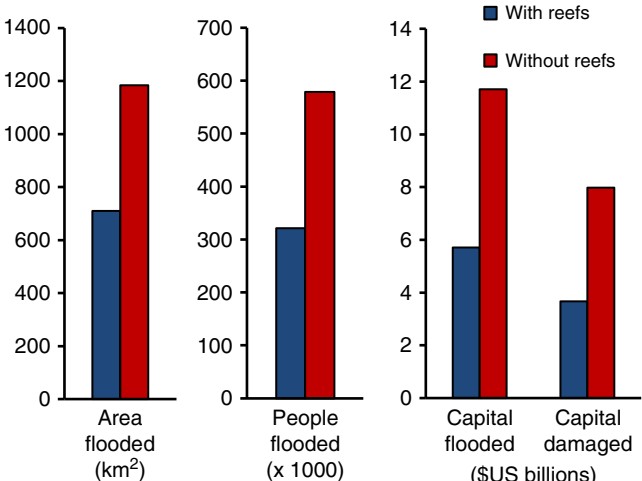

**Fig. 2** Annual expected benefits from coral reefs for flood protection. Estimates of the effects of reefs on avoided flooding to land, people, exposed capital and damaged capital. The differences between scenarios with and without reefs are avoided damages or present benefits of reefs

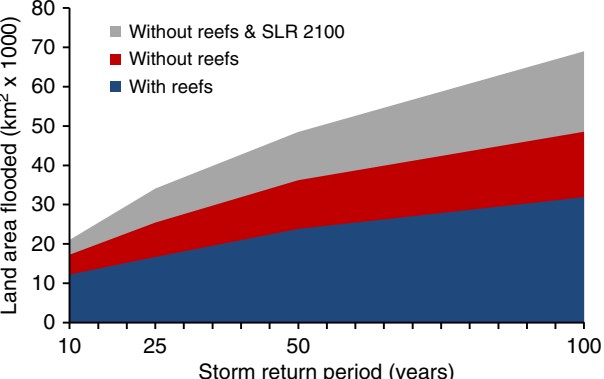

**Fig. 4** The expected land protection benefit of coral reefs at present and with sea level rise. The values are the land areas (km²) flooded globally with reefs at present, without the top 1 m of coral reefs and with relative sea level rise under a high emissions scenario (RCP 8.5 in 2100) by storm return period

**Reefs and national flood reduction benefits**. At a national scale, reefs provide annual expected benefits of hundreds of millions of dollars in avoided flood damages for five countries and millions of dollars in annual benefits for more than 20 additional countries (Table 1). Reefs also reduce annual flooding by more than 200,000 people (Fig. 2). For extreme events (e.g., 100-year events), reefs avert billions to tens of billions of dollars in damages for more than 10 countries (see Supplementary Table 1). The United States ranks among the top 10 countries that benefit from reefs (Table 1) mainly because of Puerto Rico.

The national benefits of reefs for flood protection can be considered not just in total built capital and people protected but also relative to the sizes of the national economy and population (Table 1, Supplementary Tables 1 and 2). These results highlight the importance of reefs to many smaller island nations in the Caribbean and the South Pacific, which receive significant benefits relative to their gross domestic product (GDP) (Table 1). The flood protection benefits of coral reefs are particularly critical in the Philippines, Malaysia, Cuba, and the Dominican Republic (Table 1). In these countries, reefs are important for averting damages both to built capital overall (total dollar value of national avoided losses) and relative to the size of their economies (i.e., total dollar value of national avoided losses/GDP).

**Reefs and local flood reduction benefits**. At a local scale (i.e., in 20 km shore units), we identified the critical areas that likely receive the greatest flood protection benefits from coral reefs (Fig. 5, Supplementary Fig. 1). The places where reefs avert flood damages to people are more widespread geographically (Supplementary Fig. 1), whereas the avoided damages to built capital are more concentrated near urban centers in countries such as Indonesia, Philippines, Saudi Arabia, and Mexico (Fig. 5).

## Discussion

Reefs provide significant annual flood protection savings for people and property, particularly from the most frequent storms. Annual expected damages from flooding would more than double, and costs from frequent storms would triple without reefs. These quantitative, spatially explicit analyses highlight where reefs provide the greatest flood protection services, locally, nationally, and globally. They also identify where future reef loss may have the greatest impacts and where enhanced management, conservation, and restoration will deliver the most benefits.

By integrating economic, ecologica,l and hydrodynamic models, we show variation locally, nationally and regionally around the general pattern that reefs provide the most flood protection benefits in storm belts with extensive, shallow, and rugose coral reefs; land at low elevation; and assets concentrated on the coast (Fig. 5). Importantly, flood protection is just one of the services provided by reefs, and our analyses identify benefits only from the topmost 1 m of the reef profile.

Reef flood protection benefits are particularly critical for many small island and developing States, which have a limited capacity (relative to their GDP) to respond to severe flooding and the losses of natural coastal defenses. The protection of nearshore shallow reefs should be a high priority for these nations as a

| Table 1 Countries that receive the most flood protection benefits from reefs | | | | |
|---|---|---|---|---|
| **Annual averted damages ($ millions)** | | | **Annual averted damages/GDP** | |
| 1 | Indonesia | 639 | Cayman Islands | 0.98 |
| 2 | Philippines | 590 | Belize | 0.37 |
| 3 | Malaysia | 452 | Grenada | 0.30 |
| 4 | Mexico | 452 | Cuba | 0.25 |
| 5 | Cuba | 401 | Bahamas | 0.16 |
| 6 | Saudi Arabia | 138 | Jamaica | 0.14 |
| 7 | Dom. Republic | 96 | Philippines | 0.13 |
| 8 | United States | 94 | Antigua & Barbuda | 0.13 |
| 9 | Taiwan | 61 | Dom. Republic | 0.11 |
| 10 | Jamaica | 46 | Malaysia | 0.09 |
| 11 | Vietnam | 42 | Seychelles | 0.06 |
| 12 | Myanmar | 33 | Turks & Caicos | 0.06 |
| 13 | Thailand | 32 | Guadeloupe | 0.05 |
| 14 | Bahamas | 14 | Indonesia | 0.04 |
| 15 | Belize | 9 | Solomon Islands | 0.04 |

Annual expected benefit of reefs for flood protection in terms of annual averted damages to built capital ($ millions per year) and relative to Gross Domestic Product (GDP). The values are the difference in expected damages to built capital with and without reefs

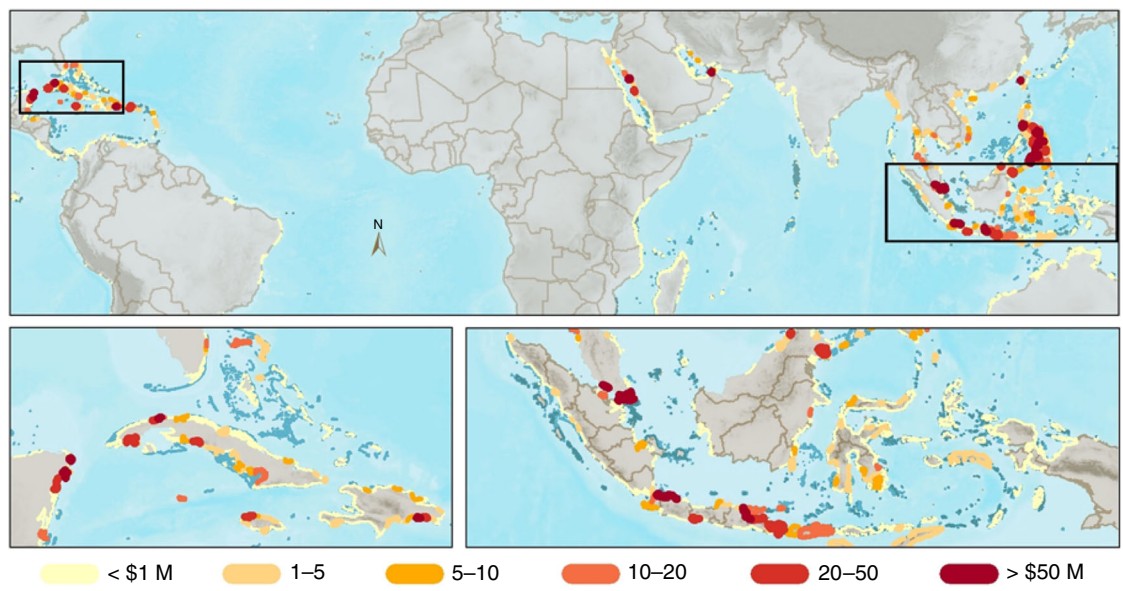

| < $1 M | 1–5 | 5–10 | 10–20 | 20–50 | > $50 M |

**Fig. 5** The value of coral reefs for flood protection. Circles represent the annual expected benefit from coral reefs for flood protection ($US millions). The values are the difference in annual expected damages with and without (the top 1 m) of reefs for the 20 km coastal study units

critical part of their coastal management and adaptation strategies. Reefs also offer indirect flood reduction benefits by reducing social vulnerability (e.g., through nutrition and livelihoods) and improving coping and adaptive capacity[12,13].

Explicit valuations of protective services are particularly critical for coral reefs as their role in flood reduction is not easily observed. Because they are below water, it is difficult to know when coral reefs have been degraded and further to make the connection between reef loss and flood damage. In contrast, the loss of intertidal habitats such as mangroves and marshes is visibly apparent, communities recognize connections between habitat loss and flood damage, and this connection has influenced large-scale restoration practices and national policies[12,40,43]. An understanding of spatial variation in flood reduction benefits is crucial as decision-makers allocate funding for risk reduction and adaptation among nations and at provincial and municipal levels[46,47].

Based on prior work and our own sensitivity analyses, the greatest sources of uncertainty in coastal flood risk assessments are estimates of topography and bathymetry. Given that flooding and damage from tropical storms are among the greatest risks to people and property, better elevation and depth data is urgently needed. Fortunately, in the past decade there has been a substantial increase in the availability of high-resolution coastal elevation data through the widespread use of LIDAR. Nearshore bathymetry, however, remains a major gap, though there are advances in remote sensing that could help[48]. Coral reef biologists and managers could address these data gaps by adding simple measures of reef height and rugosity to existing monitoring programs. These measures would improve the valuation of reef services and the assessment of flood risk, and could inform aid, insurance, and development decisions.

Our coastal flooding analyses have several significant, combined improvements over other recent global flooding analyses[2–4] including the downscaling to a 90 m resolution; consideration of hydraulic connectivity in the flooding of land; the use of 30 years of wave, surge, tide and sea level data; reconstruction of the flooding height time series and associated flood return periods[49]; and the use of country-specific adjustments to allocate GDP per person. Our global flood risk models also include ecosystems and nearshore bathymetry for the first time, which represent critical advances in the assessment of flood risk. Major remaining constraints for global coastal flooding models include the consideration of flooding as a one-dimensional process and the difficulty in representing flooding well in smaller islands.

Our estimates make a compelling case for present-day annual investments in reef management and restoration, because they are conservative. They do not assume that reefs will disappear altogether under a business as usual scenario, nor do they rely on rare, large storms. Unlike prior site-based analyses, our without reefs scenario assumes only a modest 1 m change in reef profile. Unfortunately these changes could happen quickly as there are many stressors that have and continue to contribute to the rapid loss of coral reefs[50–52]. This flattening of coral reefs has been observed globally[14,25,26] and can be accelerated by coral bleaching, as witnessed during the 2015 El Niño. In the long term, these effects could be coupled with flooding impacts from a 1 m or more rise in sea levels[3] and lead to compounding effects later in the century. However these effects are not foregone conclusions and in some areas reefs are still in good condition and even growing. The challenge will be to maintain, improve and restore healthy reefs, which will likely require more innovative effort in the areas where the protection benefits are greatest, i.e., directly adjacent to populated areas. Better decisions in coastal development could reduce risks to both people and reefs.

The economic valuation of reefs at local, national and global scales should inform the policy and practice of many agencies, businesses and organizations across development, aid, insurance, and conservation. This valuation also highlights the cost effective solutions that can be sought in reef conservation and restoration for reducing coastal risks[13,19]. Our results value coastal protection from natural infrastructure in the terms used by finance and development decision-makers (e.g., annual expected benefits) so that they can be explicitly considered alongside other common metrics for built infrastructure within national economic accounting.

The present degradation of coral reefs has significant social and economic costs. While there are significant concerns about the future of coral reefs, there is clear evidence that reefs can recover from large-scale stressors such as past El Niños and can be managed for recovery by reducing local stressors such as pollution, sedimentation and destructive fishing[50,51]. Reefs provide a substantial first line of coastal defense and should be better managed for this benefit.

## Methods

**Overview**. To estimate the role of coral reefs in coastal protection, we built on prior work that examines the effects of flooding on people and built capital across large regions[3]. To assess benefits, we follow the expected damage function (also known as the damage cost avoided) approach, which is commonly used in engineering and insurance sectors and recommended for the assessment of coastal protection services from habitats[12,38,40]. The benefits provided by reefs are assessed by their avoided flood damages. We summarize the main steps of the expected damage function approach (Fig. 1) and describe key aspects of this methodology here and in the Supplementary Methods. Define coastal profiles and study units: we delineated cross-shore profiles every 2 km for all coral reefs globally, and grouped these into 20 km study units across all coral reef coastlines (see Supplementary Fig. 3). Estimate offshore hydrodynamics: we identified sea states offshore for each profile from the combined effects of waves, astronomical tides, storm surge, and mean sea level. We used global wave and sea level numerical hindcast datasets from 1979 to 2010, which have been used extensively and validated with instrumental data[53–56]. Estimate nearshore hydrodynamics and the effects of reefs at each profile, we propagated the waves through the reef profiles, using a propagation model that accounts for shoaling, breaking and the friction induced by the coral reefs. From the wave propagation, we calculate the wave run-up on the shore[14,16,45,57,58]. Define extreme water levels along the shore: we combined run-up and sea level to estimate flood heights at the coastline[59]. We then calculated the flood heights for four storm return periods. Identify people and assets flooded: for each profile and storm return period, we identified flooding levels on land by intersecting the flood height with topography. We then developed a flood envelope across each 20 km study unit and calculated the land, people, and built capital within this envelope[2,3,60]. Develop flooding scenarios with and without reefs: we repeated the steps above for reef bathymetry under current conditions and for a reef bathymetric profile reduced by 1 m and with lower friction. Identify relative effects of climate change on flooding: we also considered the effects of climate change by comparing the land areas flooded at present, with 1 m reef loss and with reef loss and sea level rise[61] under a high emissions scenario.

**Coastal profiles and study units**. We divided the coral reef coastlines into four regions: Pacific Islands, Latin America and the Caribbean, Indian Ocean and Red Sea, and Asia and Australia (Supplementary Fig. 4). The global distribution of coral reefs has been compiled by numerous partners and most recently been updated as part of the Reefs at Risk Revisited project database[8]. We divided the coastline in to cross-shore profiles every 2 km (e.g., Supplementary Fig. 5). We aggregated the results from the 2-km coastal profiles into larger study units that were ~20 km wide (Supplementary Fig. 6).

**Offshore hydrodynamics**. The offshore hydrodynamic conditions required for the propagation models include wave climate and sea levels globally. We used different datasets: a global wave reanalysis[54]; astronomical tides[62,63]; the dynamic atmospheric correction for the meteorological residual[56]; and mean sea level data[53] compiled from historical numerical reconstructions[64] and satellite altimetry from 1979 to 2010. Storms are generally captured well in these wave data though the wave heights in some of the hurricane events can be underrepresented[54,65]. At the offshore end of each coastal profile, we identified the wave climate information (e.g, significant wave height) and sea level.

**Nearshore hydrodynamics data**. We combined topographic[66] and bathymetric data into an integrated set at each geography. We use the shuttle radar topography mission SRTM 90 m database for global elevation[66], which has been identified as

the best globally available digital elevation model and has been used in other regional and global flood models[2–4]. We used the ETOPO bathymetry[67] globally and combined it with the SeaWiFS (Sea-Viewing Wide Field-of-View Sensor) bathymetry for coral reefs, which was collected over the period 1997–2002[68,69]. Some reefs may have lost some of their height (i.e., lower bathymetric profile) in the past 10–15 years, in which case some geographies may already be seeing the increased flood risks that we predict in these analyses.

**Nearshore hydrodynamics reef wave model.** Wave propagation over the reef is calculated from linear wave theory. Wave propagation is modeled at shore-perpendicular, one-dimensional transects therefore processes such as longshore currents, are neglected. The evolution of a wavefield of root-mean square (rms) wave height $H$ with weak mean currents is computed by solving the wave energy balance equation:

$$\partial E_w C_g / \partial x = -(D_b + D_f + D_v) \tag{1}$$

where $E_w$ is the wave energy density and $C_g$ the group velocity. The dissipation of wave energy flux is caused by wave breaking ($D_b$), bottom friction ($D_f$), and the presence of vegetation in the water column ($D_v$), which is not considered in this study. Equation (1) is widely applied in coastal studies to assess wave propagation (e.g., SWAN)[70] and previously applied to reef environments[71]. $D_b$ and $D_f$ are expressed following Thornton and Guza[72]:

$$D_b = \frac{3\sqrt{\pi}}{16} \rho g \frac{B^3 \cdot f_p}{\gamma^4 h^5} H^7 \tag{2}$$

$$D_f = \frac{f_w}{16\sqrt{\pi}} \left( \frac{\sigma}{\sinh(kh)} \right)^3 H^3 \tag{3}$$

where $\rho$ is water density, $g$ the constant of gravity, $k$ the wave number, $\sigma$ the angular wave frequency and $f_p$ the peak frequency. The breaking coefficient $B$ and breaker index $\gamma$ have the default values of 1.0 and 0.78 and the bottom friction coefficient $f_w$ is taken as 0.01 for sand beds[57,72].

In our model, we implement recent studies on wave transformation by coral reefs[16,73] and replace the breaker index ($\gamma$) by an expression where $h/H$ provides the relationship between water depth and wave height at breaking conditions:

$$\gamma_{coral} = 0.23 \tanh \left[ 2.3143 \left( 1.4 - \frac{h}{H} \right) + 3.6522 \right] 0 < \frac{h}{H} < 2.8 \tag{4}$$

**Nearshore hydrodynamics total water level model.** The total water level (i.e., flood height) along shorelines is a function of mean sea level, astronomical tide, storm surge, and the run-up of waves[53]. The run-up represents the wave-induced motion of the water's edge across the shoreline and is built of two contributions, namely the wave setup at the shoreline and the swash representing oscillations about the setup. The run-up calculation requires obtaining the local wave conditions at the shoreline using the reef wave model above.

**Nearshore hydrodynamics computation of wave setup.** The wave-setup is obtained from the conservation of mass and the momentum equations[74]. In our one-dimensional setting, the computation of the wave-induce setup is based on the vertically integrated momentum balance equation[75]. Similar implementations have been used in previous work to evaluate the effect of vegetation on wave-induced setup[45] and in coral reef environments[76].

**Nearshore hydrodynamics computation of wave run-up.** The 2% exceedance level of wave run-up maxima generated by random wave fields on open coast sandy beaches was estimated in Stockdon et al[77] as:

$$R_{u\_Stockdon} = 1.1 \left( 0.35m\sqrt{H_0 L_0} + \frac{\sqrt{0.004H_0 L_0 + 0.563 H_0 L_0 m^2}}{2} \right) \tag{5}$$

where $H_0$ is the offshore significant wave height, $L_0$ represents the deep-water wave length, $T_p$ the peak period, and $m$ the bathymetry slope in the foreshore beach slope. This equation expresses run-up as a function of empirical estimates of incident wave setup at the shoreline (first term of the equation) and the swash incident and infragravity band frequency components (second term of the equation). In this analysis, the first term is replaced by the calculation of the setup contribution as explained under nearshore hydrodynamics computation of wave setup above.

For the swash and infragravity band frequency components, $H_0$, $L_0$, and the foreshore slope $m$ must be determined. Using the wave propagation model over the reef, we calculate the breaking point position ($x_b$), breaking depth ($h_b$), and breaking height ($H_b$). Then, following Stockdon et al. we deshoal the wave height to deep water to obtain $H_0$. $L_0$ is calculated for the corresponding peak period ($T_p$).

The foreshore slope $m$ is obtained for each of the different profiles from the DIVA-GIS dataset (http://www.diva-gis.org/).

In our approach, we assume that Stockdon et al.[77] can be applied to coral reefs as the model was developed to include barred beaches, which resemble coral reef protected beaches. Modifications of the same formula have been applied previously to estimate the effect of vegetated ecosystems on run-up[45].

The application of a one-dimensional model neglects some of the hydrodynamics that occur on natural reefs, such as longshore flow and lagoon circulation. However, this 1-D approach is common in reef studies, either with the same wave action balance equation used here or in more complex numerical hydrodynamic studies[14,15,41,45,71,78]. Flood models based on the wave action balance equation are widely employed for coastal modeling[79]. The consideration of non-linear effects is only possible using phase resolving models (e.g., XBeach) at local scales (e.g., bays)[12,15,80,81]. This modeling approach is not feasible at the global scale because of computational capacity and the lack of high-resolution bathymetric data and especially if risk is to be evaluated probabilistically. We have shown that the wave propagation approach in our global reef flooding model performs very well when considered against the results of one of these phase resolving models (see Supplementary Fig. 7). The changes in flooding in our global model also are consistent with changes observed in a site-validated, XBeach model that also considers flooding with changes in reef friction and sea level[15]. In Supplementary Data 1, we summarize the models, equations, and assumptions that we used in our global model and compared their benefits and limitations relative to approaches that are feasible in local or smaller scale studies.

**Extreme water levels and flood height reconstruction.** From the propagations of waves and the calculation of total water levels onshore (above), the reconstruction of the flood height time series at the most onshore points is based on multi-dimensional interpolation techniques[59]. We apply a peak over threshold method to select extreme flood heights and fit a general extreme value distribution[82] to obtain the flood heights associated with the 10-, 25-, 50- and 100-year return periods. The methodology has been tested in case studies and validated with observations[83,84].

**Estimating reef benefits.** To examine the current value of reefs for coastal protection, we compared flooding under current conditions, "with reef", to the flooding in a scenario "without reefs". In our "without reefs" scenario, we do not assume the loss of the entire reef habitat; we assume only the loss of the top 1 m in height across the reef bathymetric profile. Many ecosystem service assessments assume the entire loss of a habitat for estimating benefits. For example, the replacement cost method, which is the most commonly used method for estimating the benefits from mangrove and reef habitats[12], identifies the flood reduction benefits from habitats by estimating the cost of replacing them with seawalls or breakwaters. Many problems have been identified with the replacement method and it provides estimates of values ten times higher than the recommended expected damage function approach that we follow[12,40].

The without reefs scenario is not meant to be a prediction of site-specific trajectories for reefs, but nonetheless this level of loss is already observed to be happening in many places[14,25,26] and is conservative relative to future predictions of reef loss[28–30]. In addition to the widely observed declines in coral cover, growth and condition, all of which affect reef height[20,21], new measures of seafloor elevation show that bioerosion and carbonate dissolution are degrading height across all reef habitats including on reef flats[26]. Damage from storm events can also create losses in reef height of 1–3 m[85,86] and can devastate whole shallow reef frameworks[87]. Past storms have removed many branching and massive corals at the shallowest depths[24,85]. Shallow corals have evolved with intermittent storms and can recover from them, but this is more difficult when reefs are exposed to multiple stressors[24].

We developed regional friction factors, $f_w$, following Sheppard and others[14], who examined the relationship between percent of live coral cover and friction (Supplementary Table 3). Based on the available literature[88], we used different friction coefficients for each of the four major study regions. Given broad estimates of coral condition[8], we assumed current condition was best in Micronesia ($f_w = 0.20$); lower in the Indian and Indo-west Pacific regions ($f_w = 0.16$); and lowest in the Caribbean ($f_w = 0.14$). Assuming a loss of the living coral cover, we then estimated friction to be 0.08 with reef loss[14] in all four regions.

**Calculating people and assets flooded.** We assessed flood heights along each coastal profile and then identified the area flooded within each coastal study unit. We extended the flood heights inland by ensuring hydraulic connectivity between points at a 90 m resolution; a significant advance over more common bathtub approaches in earlier global flooding models. From the flooding levels and flsooding extent, we calculated the total area of land affected and damages at each study unit. Flooding maps were also intersected with population data[60] after resampling from the original 1 km resolution to the 90 m of the digital elevation model. Existing artificial defenses such as seawalls were not assessed, because data on defenses only exist for a very few areas globally; these built defenses are also less common in tropical, developing nations.

We expanded on earlier approaches to infer built capital from population data[2,3] by identifying the ratio between built capital per capita and the gross domestic product (GDP) per capita for each country[3] in 2011 US$ using

information from the World Bank[89]. We filled data gaps for several countries by using the average from countries with similar income levels and affiliation to the Organization for Economic Cooperation and Development (OECD). The overall global mean ratio that we obtained (2.67) is similar to that obtained by Hallegatte and others (2.8)[2]. However, we did identify significant differences in the ratios across some countries and regions (e.g., Cuba—4.53, Vietnam—3.22, Australia—3.17, Philippines—2.68, United Arab Emirates—1.98, Micronesia—1.38).

**Assessing damages and estimating annual benefits**. We followed existing approaches for assessing the damages to built capital as a function of the flooding level[4]. We calculated the percentage of built capital that has been damaged ($D$) for a given flooding level $h$ and a certain coefficient $k$ that must be calibrated as $D(h) = h/(h + k)$. This curve indicates that as flooding level increases, the percent of damages to built capital also increases. While there is debate about the right $k$ to choose, we have followed others in using $k = 0.5$[4], which means that the built capital flooded at 1 m of depth loses 50% of its value. We follow standard terminology where the total built capital flooded is the exposure of assets and the value lost is the damages. The economic benefits of flood protection are the avoided damages.

In addition to assessing risk and damages for particular events (e.g., 100-year storm event), we also examined average annual expected loss[90]. To estimate annual risk, we integrated the values under the curve that compares built capital damaged by storm return period, i.e., the integration of the expected damage by the probability of the storm events[4].

**Sea level rise**. We assessed the potential added impacts of sea level rise and reef loss by considering the additional land area flooded in 2100 by storm return period under a business-as-usual (high) emissions scenario, representative concentration pathway (RCP) 8.5. To estimate the added effects of sea level rise and reef loss, we recalculated the flood heights at every cross-section worldwide considering all the prior factors and adding the local relative sea level rise projected in Slangen et al.[61] for RCP 8.5 by 2100 (see Supplementary Information for more details). Once the projected flood heights were calculated, the assessment of flooding level and total area of land affected followed the same approach as above.

**Sensitivity analyses**. We conducted extensive sensitivity analyses for parameters across the models and find that the results are robust to changes in the key parameters across their natural ranges of variability. These tests are summarized here and described more fully in the Supplementary Information. The flooding model is the most critical model for estimating flood heights. After testing all parameters in the model, we identified that the estimates for reef friction, water depth, and the wave breaking parameters were the main ones affecting the run-up contribution to flood heights. We examined the effects of changes in these parameters by incrementally changing them across their range of variability and running tens of thousands of profiles across different reef types (Supplementary Table 4). In sum, changes in the estimates of the friction ($C_f = 0.08$–$0.20$) and wave breaking ($\gamma_{coral} = 0.2$–$0.6$) parameters have only small effects with only approximately 10% changes in run-up from the minimum to maximum of these parameter estimates. Changes in water depth from 0.1–1 m had the largest effects on the results. Each 10 cm change in depth changed the run-up contribution by ~2%. Additional uncertainties of input data such as digital elevation models or population data on global flooding models have been discussed by Hinkel et al.[4]

We also did sensitivity analyses on the damage function model with other parametrizations of $k$ ($k = 0.2, 0.3, 0.4$). Lowering $k$ lowers the total value of the built capital damaged, but has little effect on the relative effectiveness (% difference) of reefs for risk reduction. Lower $k$ values slightly increase the relative effectiveness of reefs making our use of $k = 0.5$ the most conservative for comparisons.

**Data availability**. All results are mappable and downloadable at http://maps.oceanwealth.org/. The underlying data sets including Global Waves and the Python source codes for key analyses are available on request from IHCantabria at ihdata@ihcantabria.com.

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

## Acknowledgements

We gratefully acknowledge support from the World Bank Wealth Accounting and Valuation of Ecosystems (WAVES) Program, the Lyda Hill Foundation, Science for Nature and People Partnership, Lloyd's Tercentenary Research Foundation, a Pew Fellowship in Marine Conservation to MWB, the German International Climate Initiative (IKI) of the Federal Ministry for the Environment, Nature Conservation and Nuclear Safety (BMU) and the Spanish Ministry of Economy and Innovation (BIA2014-59718-R). Thanks to L. Hale, A. White, S. Narayan, and D. Trespalacios for comments on earlier versions and to L. Flessner and D. Trespalacios with help on figures.

## Author contributions

M.W.B., I.J.L., and B.G.R. conceived the study; I.J.L., B.G.R., P.M., P.D.-S., and F.F. developed and implemented the analysis; M.W.B. led the manuscript writing; and all authors made substantive contributions to the text.

## Additional information

**Competing interests:** The authors declare no competing interests.

