## [Peer Review File · Nature Communications]

Reviewers' comments:

Reviewer #1 (Remarks to the Author):

Thanks for the opportunity to review “the global flood protection savings provided by coral reefs.”

The paper presents a very impressive study to assess coastal risks in all coastline segments with coral reefs, and measure the increase in risks if these coral reefs lost one meter, due to various environmental impacts. They find a significant increase in risks, with a cost larger than \$100 billion for the 100-year event.

I find the paper really interesting, it covers a really important topic, and represents an impressive effort to bring together multiple disciplines, from wave energy modeling to economic losses modeling. I think these results really need to be published as they have a great research and operational interest.

I have a few comments that – I think – should be taken into account before the paper is published.

First, the paper is not rigorous on the language and how results are presented. It makes the text confusing, and reduces the potential impact of the paper. There are many examples where I think the authors use the wrong words:

- Line 102 “reduced annual flooding for more than 200,000 people” – my understanding is that 200,000 fewer people are flooded on average, which is very different than what the paper says. Maybe the authors could replace by “reduced annual flooding by more than 200,000 people”
- Line 103 and Table S1: there is confusion between exposure and losses (in many places in the paper). The caption of Table S1 says “protection benefits [...] in terms of exposure of built capital to flooding” so it is the exposure, not the “protection benefits” as in line 103. Unless Table S1 shows the avoided losses (in which case the caption needs to be rewritten).
- The wording “built capital that has been damaged” or “built capital damaged” (line 314, 348, Y-axis label of Figure 3) is really unclear. For me, the damaged capital is the exposure (if a \$1m

building suffer losses for \$50k, then the built capital that has been damaged is \$1m and the build capital damages are \$50k). In the paper, the reader never really knows if the authors talk about exposure or losses.

- Figure 3 caption says that the figure shows benefits while it shows the losses due to floods.
- The authors have results for all return periods (Figure 3) but they use mostly the 100-yr return period results, which is confusing. In the abstract, for instance, one would like to know the impact of the reefs on average losses (based on Table 1). Figure 3 suggests that most of the effect of reefs on annual average losses will come from the frequent events (return period less than 25 years), and this would be important to flag too...

I think the authors should go through their paper carefully to ensure they use a more precise language and avoid confusion.

Second, I'm curious about the impact that flood protection infrastructure would have on the estimates. Seawalls are not included in the type of topographical information they use, but they can avoid coastal floods and they play a key role in the assessment of flood risks. To assess the impact of annual average losses, line 319-322, the authors would need to have a catalogue of coastal protection for the coastal segments they consider... And in Figure 3, the losses probably go to zero for low return period thanks to protection (even non-reinforced coastline are often assumed to provide protection against the 2-yr return period event).

This limit should not affect their assessment for the 100-yr flood, since it seems realistic that the considered coastline does not benefit from a protection beyond the 100-yr event. In short, the paper could be clearer on the role of artificial protection and mention this limit upfront (including in the abstract and Table 1). Another way of evaluating the value of coral reefs through flood protection would be to assess the cost of building infrastructure to reduce the risk by the same amount (even though such infrastructure would deliver much less benefits in other dimensions such as food security, biodiversity, or recreation).

Third, the paper says that a storm can create 1-3m loss of reef (line 284). I guess it mean that the protection provided by the dead coral reef (the one with a 1m loss) would work only once. It seems to me that this is a critical point that is much more important than the difference between current reefs and reefs reduced by 1m – the main difference is that a reef reduced by 1m is dead and therefore will disappear after one or two or three storms, then providing absolutely no protection. If

I'm right, then the main economic cost of dead reefs is not included in the analysis. To get to that, it would be useful to also run the model in the absence of coral reef.

There are also a couple of details. There are a few issues with the equations: line 250, x is not defined (it is defined in the SI). Line 267, T_p does not appear in the equation (also in the SI). Also, the authors should refrain from using acronyms when possible. I think they do not need to replace "flood height" by "FH" – it makes the paper more difficult to read.

Finally, I have to say that I'm not a specialist in hydrological modeling, so I cannot assess the validity of the modeling assumptions on flood extension in the paper.

To conclude, I think the paper makes an important point and that a global analysis like this one has a great value – I hope this paper can be published, after the little issues that I flagged are corrected.

Reviewer #2 (Remarks to the Author):

This manuscript attempts to establish the savings in flood protection afforded by coral reefs. The study is another attempt to ascribe monetary value to one of the ecosystem services provided by coral reefs to highlight their importance. Similar attempts have been attempted for more than a decade including attempts by the World Bank in 2005. The study also attempts to provide a global overview of flood protection savings. While the aim is laudable I believe the results are not novel. I have serious reservations concerning the gross assumptions made in the study and flaws in logic of reef systems that these expose.

1. The assumption made is that globally all reefs will lose 1m of reef due to storms and other impacts. There are several concerns related to this assumption:

a. There is a paucity of literature to back up this assumption. Alvarez et al. (2009) is cited as the main reference to support such changes. This study was limited to the Caribbean only and examined different reef locations. Such conclusions cannot and should not be transferred to other reef systems globally, which has responded very differently to Caribbean reefs over the past century.

Indeed, there are many more studies that have shown that reefs have grown in the past century – why not use those scenarios?

b. The assumption above also assumes that all reefs have a 1m thick veneer of living coral. This assumption is badly flawed. The manuscript does indicate that removal of the reef framework is not considered in the model. This admission undermines the contention that 1m of reef can be removed from all reefs globally. For example, the majority of Pacific reef systems have only a very thin veneer of living coral (<0.3m). Furthermore, most Pacific and open ocean reef systems do not have coral on the reef rim. Rather they are colonised by coralline algae (which can grow higher in elevation than coral) due to the high nature of the reef rims.

c. The critical zone for wave breaking and reduction of wave energy is the upper reef crest and reef flat. The literature cited regarding reef development generally does not refer to these zones. Indo-Pacific reefs are generally depauperate in coral on these surfaces (or have an extremely patchy and thin veneer). Consequently, removal of living coral is unlikely to change the elevation of the reef crest and reef flats substantially in these zones. Given the Indo-Pacific accounts for the vast majority of reef globally, the assumption of the study is invalid. The loss of 1m of coral from the forereef slope will also have minimal impact on wave transformation processes.

While the authors may argue the actual reef responses are irrelevant to the point being made that reefs provide flood protection I disagree. It is irresponsible to set up an academic argument founded on false assumptions. In general the manuscript lacks convincing insights into reef development and change, or reef hydrodynamics and without such a foundation the economic analyses really have no value.

2. The hydrodynamic analysis is also simplistic and based on limited studies. What is not reflected in the study is that substantial changes in water depth across reefs fundamentally changes the types of waves and wave frequencies impacting shorelines. These changes radically alter the types of waves and nature of flooding of coastal lowlands. The manuscript fails to recognise these critical differences. The manuscript uses outdated coastal engineering tools to assess the flooding.

3. The manuscript also assumes that the coastal margin abutting reef systems is inert and will simply be drowned with increasing water level. In essence elevated water levels simply flood the coast to a different bathymetric contour line in the often used 'bathtub' approach. Such an approach has been widely criticised as it fails to reflect the dynamic nature of coastal margins. Indeed, numerous studies have shown that coastal margins are dynamic and can change their elevation with respect to high runup levels.

Collectively, the detailed analysis stacks up multiple gross and incorrect assumptions that render the analysis of flooding meaningless. Consequently, the results and discussion sections are full with generic and over-extended assertions of the magnitude of coastal flooding, likely impacts and their future changes.

Currently, the desire to make global assertions of impacts is simply not believable on the basis of the assumptions and evidence presented. I would have thought a better approach may have been to use more detailed analysis of selected sites where losses have been documented. Application of latest numerical modelling approaches could then highlight precisely how flood wave processes will change and influence coastal hinterland. Similarly, this would have allowed the article to explore what savings are made at sites where the reef has been accreting over the past century.

On the basis of the above comments I believe the manuscript should be rejected at this time.

Reviewer #3 (Remarks to the Author):

The manuscript submitted by Beck et al. is a really interesting and important contribution. They demonstrate the risk reduction benefits of coral ecosystems at a global scale, using process-based methods. They also provide outputs in units that will resonate with different readers or stakeholders. The information provided will be useful to coastal managers, government officials, engineers and scientists alike. However, although the authors seemed to be very careful in their analysis, I found the paper hard to follow at times, both in its style, but also because of lack of clarity on in many of the paper's sections.

I suggest that during their revision, the authors spend time improving the overall style and readability of the manuscript. Special attention should be paid to the use of words, transitions between sentences and paragraph, paragraph structure, and overall flow of information. For example, in the paragraph starting in Line 95, where the first sentence talks about future sea level rise, the second refers reef elevation loss under current conditions, and the third refers to sea level rise. This strange arrangement led to some confusion in my reading of the manuscript, and so can potentially confuse other readers.

Another source of confusion is the loose uses of the words climate change and flooding. For example, in the Abstract, the manuscript reads "Climate change will increase flood risk [...]" (Line 24). This sentence is not clearly linked to the previous one, which was discussing countries with most to gain from reef conservation. But it is also unclear why the authors say that climate change will increase flood risk: is it because of increased rainfall, higher waves, stronger hurricanes, sea-level rise? Similarly, in the first paragraph, sentences 2 and 3 (Line 34) don't really follow each other, and once again the term "climate change" is used loosely. It is left to the reader to guess which climate

chance impact will increase flooding. The same observations are valid for the use of the word “flooding”. To most readers, “flooding” usually means that water elevation at a particular location is higher than usual, and usually it is caused by intense rain, or by a storm surge. Here, most of the flooding that is reduced by corals is due to a reduction in wave runup, which is one portion, and generally not the most important, of flooding.

So I suggest that authors pay attention to readability of their manuscript, and take the time to define as early as possible how different terms are used in the manuscript, such as flooding or climate change. Other stylistic comments, which are not exhaustive:

1. Please revise the use of the word “underwater” in Line 64. That sentence is not clear.
2. Are there “a couple” (i.e., two) national studies (Line 64) to quantify ecosystem benefits or “at least a couple”?
3. Expand on what “other impacts” are causing loss of coral structure (Line 80). Are human activities part of these causes? This might be important for intended audience to know.
4. Replace “In” by “For” when describing results for 25 and 100 year events (Lines 90 and 92)
5. Check use of the word “currently” in line 90.
6. On Line 137, instead of “reduce wave energy dissipation”, why not say “increase transmitted wave energy” so it can be understood by wider audience?
7. On line 142, the expression “reducing flooded areas” is confusing. Please be more clear.
8. What are Small Island Developing States (Line 147-148)? Are they the same as the nations discussed earlier (Line 119)? Please clarify.
9. The word “business as usual” is usually used in climate projection. Maybe use another word? (Line 152)
10. Provide a reference for the claim that 1 m of reef height can happen in the near term (Line 154). Also, this sentence give a sense of urgency that seems out of place. Should it be in the motivation section of the introduction?
11. The last two paragraphs of the Discussion seem out of place. Please review for readability.

In addition to these general comments, I suggest that the authors review their manuscript and pay attention to how they present information. I feel that the Introduction does not properly motivate the question being researched, and the discussion section feels unstructured in general. I suggest that the author revise the order of their arguments and the points and connections they want to make. Other technical comments below:

1. The introduction does not clearly explain why it is important to look at how much coral reefs reduce flooding caused by wave runup. The authors mention coral reefs in the first paragraph of the introduction (Line 36), then again later in paragraph before last. However, it is not revealed until the

last sentence why it is important to look at the importance of reefs in risk reduction, which is that reefs, globally, are losing rugosity, elevation and will be more at risk of degradation as time passes. I suggest that the authors motivate this study as early as possible, probably in the first or second paragraph

2. It is unclear how modeling scenarios were created. They are described in the last paragraph with very little information to justify them. Information about the flattening of reefs is provided after the scenarios are presented, and the reference for the flattening is only for the Caribbean. What about the other regions of the world? Also, in that last paragraph of the introduction, the authors mention a 1-meter decline in reef height, but this value does not seem to be justified, and it is unclear if it applies in the future or in the present. I suggest that the authors spend more time carefully explain the rationale behind their scenario, then explain clearly what these scenarios are.

3. Does the reef distance to shore matter? The authors did not provide any information about the types of reefs that provided more protection than others, and why. This might be interesting for those trying to understand what matters most in reef restoration.

4. In sentence starting on Line 110, the authors claim that “hotspots areas” suffer many storms. Can they comment on the ability of the reefs to withstand these storms and recover in between storms so they can deliver the same level of protection through time?

5. I mentioned this above, but it is important that authors clearly explain how reefs reduce flood levels, namely, that they reduce wave runup, in order to clear set expectations in readers. The first sentence of the discussion (Line 128) could be interpreted as reefs reduce storm surge levels to fast readers.

6. The authors discuss stressors to corals reefs at the beginning of the Discussion section (Line 133), then again in the third paragraph of that section (Line 146), then discuss other ecosystem services on line 179. Maybe move this type information to the introduction to help motivate the manuscript, or at the end of the Discussion section?

7. On Line 137, the authors mention that impacts on reefs can leave smooth limestone structures. It is my understanding that those consequences can be observed decades after impacts started. If that’s the case, this statement adds to my confusion as to the motivation and interpretation of the scenarios: are we looking at current conditions, future conditions with 1 m sea-level rise and loss of rugosity, or future conditions with loss of rugosity without sea-level rise? My apparent confusion argues for more clarity in the introduction on what is exactly measured and why.

8. How is the 1 m height decrease exactly modeled? The text suggests that bathymetry is lowered by 1 m (Line 77), but Figure 1, Panel B, seems to suggest that the bathymetry remains unchanged, but the friction coefficient is reduced. Please clarify and be consistent throughout the text.

9. On Lines 138-140, the authors mention, once again, that corals are flattening in the Caribbean, but they use the same reference (ref. 38) to mention that reefs are declining in height in other parts of the world. I suggest that the authors complete a more exhaustive literature survey on the subject and find references for regions in other parts of the world, or focus solely on the Caribbean.

10. The authors mention that their estimates are conservative, on Line 152. Please explain, because the model has many sources of uncertainty (water depth, water elevation, friction coefficients, etc.)
11. The discussion section does not address sufficiently the limitations of the study, and the how to interpret results properly. The methods, although sound, are not precise enough to correctly capture all the nearshore processes that affect inundation, so results are a 1st estimate, and cannot be used to make important decision. Since the authors recommend that nations use their results in their general accounting, it is important to be clear on the limitations of these results.
12. In the Nearshore Hydrodynamics Methods section (Line 244) please clarify that this is a 1-D model, and clearly state its limitations. Also provide some information about the formulation of D_f .
13. The authors do not provide information on the accuracy of their topography data (Line 255-260). This might be important to put results into perspective, especially in light of the results of their sensitivity analysis.
14. The authors do not provide sufficient information on the modeling of storm surge (Line 262) in their estimation of flood elevation. This is important as storm surge is an important component of flooding. Even in the SI, the authors do not explain how they treat the storm surge component of their dataset. The references that they provide are all for waves (SI Lines 170). This comment again argues for more
15. The runup expression derived by Stockdon is only valid for sandy beaches (Line 265-266), on relatively exposed coasts. It contains an estimate of swash and setup generated by infragravity and incident waves. In their manuscript the authors assume, without justification, that reefs affect both incident and infragravity wave processes, and that all coasts are sandy, exposed, beaches. Also, the authors use the nearshore slope (Line 268), whereas the equation asks for the foreshore slope, which might lead to large errors, especially given the general lack of accuracy of nearshore depths in global bathymetry dataset. This is a stretch of the runup formulation provided by Stockdon, and the authors do not provide sufficient information on the computation of runup, and do not justify the limitations of their approach.
16. In their application of the runup equation, the authors do no properly explain how they estimate the equivalent deepwater wave (Line 267). I suggest improving the content of the manuscript using some of the language used in the SI (SI Line 116)
17. In their application of the runup equation, the authors seem to assume that there is always a wave setup generated at the coast. However, studies by Lowe et al (2010) suggest that setup, and thus runup, shoreward of reefs is a complex function of reef geometry. Please clarify the limitations of the existing study.
18. I found the explanation of the choice of different friction coefficients for reefs, and their use in the different scenarios lacking (Lines 288-295, SI Lines 150-154). In particular, no justification was given for the choice of $C_f=0.08$ for reefs without living corals (Line 295, no mention in the SI). As mentioned earlier, I suggest that the authors clarify their methods how they treat the different scenarios.

19. I appreciate the authors did a sensitivity analysis. I suggest including nearshore slope in their parameters spaces, given that they did not apply the runup equation correctly. I also suggest that they use consistent language in the SI to compare results of their analysis (SI Lines 221-228).

20. The horizontal axis or text of Figure S3 is difficult to understand. I suggest the authors improve it.

21. The regions north of New Zealand do not seem to be included in the analysis (Figure S4). Can you clarify why?

Reviewers' comments:

Reviewer #1 (Remarks to the Author):

Thanks for the opportunity to review “the global flood protection savings provided by coral reefs.”

The paper presents a very impressive study to assess coastal risks in all coastline segments with coral reefs, and measure the increase in risks if these coral reefs lost one meter, due to various environmental impacts. They find a significant increase in risks, with a cost larger than \$100 billion for the 100-year event.

I find the paper really interesting, it covers a really important topic, and represents an impressive effort to bring together multiple disciplines, from wave energy modeling to economic losses modeling. I think these results really need to be published as they have a great research and operational interest.

I have a few comments that – I think – should be taken into account before the paper is published.

First, the paper is not rigorous on the language and how results are presented. It makes the text confusing, and reduces the potential impact of the paper. There are many examples where I think the authors use the wrong words:

We have tightened the language in the paper. In particular we have clarified that we follow standard approaches in measuring the benefits of reef protection as avoided flood damages, and we have used fewer and more consistent terms on this point.

- Line 102 “reduced annual flooding for more than 200,000 people” – my understanding is that 200,000 fewer people are flooded on average, which is very different than what the paper says. Maybe the authors could replace by “reduced annual flooding by more than 200,000 people”

We have used the authors suggested text.

- Line 103 and Table S1: there is confusion between exposure and losses (in many places in the paper). The caption of Table S1 says “protection benefits [...] in terms of exposure of built capital to flooding” so it is the exposure, not the “protection benefits” as in line 103. Unless Table S1 shows the avoided losses (in which case the caption needs to be rewritten).

The reviewer is correct. We have clarified throughout the main text and supplementary material that the main evaluation of protection benefits is the avoided flood damages. We have changed the caption in Table S1 to reflect this fact.

- The wording “built capital that has been damaged” or “built capital damaged” (line 314, 348, Y-axis label of Figure 3) is really unclear. For me, the damaged capital is the exposure (if a \$1m building suffer losses for \$50k, then the built capital that has been damaged is \$1m and the build capital damages are \$50k). In the paper, the reader never really knows if the authors talk about exposure or losses.

We have clarified throughout the mss that the main flood protection benefits are the avoided flood damages.

In the reviewers example above the built capital flooded or exposed is \$1M and the built capital damaged is \$50k. The benefits provided by the reef are the avoided damages (i.e., averted losses).

We have clarified this point as follows on Line 337- “We follow standard terminology where the total built capital flooded is the exposure of assets and the value lost is the damages. The economic benefits of flood protection are the avoided damages.”

- Figure 3 caption says that the figure shows benefits while it shows the losses due to floods. *We have clarified throughout that the main flood protection benefits are the avoided flood damages. We have generally tried to use fewer terms and focused mainly on using the terms “protection benefits” and “avoided damages”. Avoided losses and avoided damages are synonymous.*

- The authors have results for all return periods (Figure 3) but they use mostly the 100-yr return period results, which is confusing. In the abstract, for instance, one would like to know the impact of the reefs on average losses (based on Table 1). Figure 3 suggests that most of the effect of reefs on annual average losses will come from the frequent events (return period less than 25 years), and this would be important to flag too...

*Throughout the paper and in the abstract, the main results that we discuss are the **annual** expected benefits (Table 1 and Figure 2 & 5). We then also note the benefits that reefs provide in avoided damages from 10-, 25-, 50-, 100-yr return period events (Figures 3 and 4).*

For example on Line 341 we indicate that “In addition to assessing risk and damages for particular events (e.g., 100-year storm event), we also examined average annual expected loss⁸⁰. To estimate annual risk, we integrated the values under the curve that compares built capital damaged by storm return period, i.e., the integration of the expected damage by the probability of the storm events⁶.”

That is, the annual expected benefit is the area of the red polygon in figures 3 and 4.

I think the authors should go through their paper carefully to ensure they use a more precise language and avoid confusion.

We agree and we have done so.

Second, I'm curious about the impact that flood protection infrastructure would have on the estimates. Seawalls are not included in the type of topographical information they use, but they can avoid coastal floods and they play a key role in the assessment of flood risks. To assess the impact of annual average

losses, line 319-322, the authors would need to have a catalogue of coastal protection for the coastal segments they consider... And in Figure 3, the losses probably go to zero for low return period thanks to protection (even non-reinforced coastline are often assumed to provide protection against the 2-yr return period event).

We agree with many of these points but the data simply do not exist and none of the global flood models including those recently published in Nature and other high profile journals directly consider existing artificial defenses (e.g., Hallegatte 2013 NCC, Hinkel et al 2014 PNAS).

In the methods, we note on Line 319- “Existing artificial defenses such as seawalls were not assessed, because data on defenses only exist for a very few areas globally; these built defenses are also less common in tropical, developing nations.”

I recently discussed this issue with the science chiefs of the top two global firms that do modeling across the insurance industry (RMS, AIR), and they don't have access to these artificial defense data, which even when collected are treated as sensitive information by many governments.

This limit should not affect their assessment for the 100-yr flood, since it seems realistic that the considered coastline does not benefit from a protection beyond the 100-yr event. In short, the paper could be clearer on the role of artificial protection and mention this limit upfront (including in the abstract and Table 1). Another way of evaluating the value of coral reefs through flood protection would be to assess the cost of building infrastructure to reduce the risk by the same amount (even though such infrastructure would deliver much less benefits in other dimensions such as food security, biodiversity, or recreation).

As noted above, we have clarified that the data do not exist to consider artificial defenses in tropical (or temperate) areas. We have also indicated in the mss that Replacement Cost methods are possible but the Expected Damage Function approach is preferred. In fact we note that previous studies by Barbier and colleagues have shown that Replacement Cost approaches value benefits 10 times higher than Expected Damage Function approaches. Further these approaches generally assume that habitats are lost entirely (not just the top 1m) and that site-specific values of project costs can be transferred across geographies.

Third, the paper says that a storm can create 1-3m loss of reef (line 284). I guess it mean that the protection provided by the dead coral reef (the one with a 1m loss) would work only once. It seems to me that this is a critical point that is much more important than the difference between current reefs and reefs reduced by 1m – the main difference is that a reef reduced by 1m is dead and therefore will disappear after one or two or three storms, then providing absolutely no protection. If I'm right, then the main economic cost of dead reefs is not included in the analysis. To get to that, it would be useful to also run the model in the absence of coral reef.

First and as noted further below we have clarified that

Line 286- “Estimating reef benefits. To examine the current value of reefs for coastal protection, we compared flooding under current conditions, “with reef”, to the flooding in a scenario “without reefs”. In our “without reefs” scenario, we do not assume the loss of the entire reef habitat; we assume only the loss of the top 1m in height across the reef bathymetric profile. Many ecosystem service assessments assume the entire loss of a habitat for estimating benefits. For example, the replacement cost method, which is the most commonly used method for estimating the benefits from mangrove and reef habitats¹², identifies the flood reduction benefits from habitats by estimating the cost of replacing them with seawalls or breakwaters. Many problems have been identified with the replacement method and it provides estimates of values 10 times higher than the recommended Expected Damage Function approach that we follow^{12,40}.”

Second most nearshore reefs exist – indeed many corals have adapted to exist- in cyclone belts. They regularly experience significant storm damages and then over a period of 3-10+ years regrow, if they are not stressed by other impacts (see Puotinen et al. 2016, Scientific Reports).

We note that (line 300)- “Damage from storm events can also create losses in reef height of 1-3 m^{75,76} and can devastate whole shallow reef frameworks⁷⁷. Past storms have removed many branching and massive corals at the shallowest depths^{24,75}. Shallow corals have evolved with intermittent storms and can recover from them, but this is more difficult when reefs are exposed to multiple stressors²⁴.”

There are also a couple of details. There are a few issues with the equations: line 250, x is not defined (it is defined in the SI). Line 267, Tp does not appear in the equation (also in the SI). Also, the authors should refrain from using acronyms when possible. I think they do not need to replace “flood height” by “FH” – it makes the paper more difficult to read.

Agreed. We have made these changes.

Finally, I have to say that I’m not a specialist in hydrological modeling, so I cannot assess the validity of the modeling assumptions on flood extension in the paper.

To conclude, I think the paper makes an important point and that a global analysis like this one has a great value – I hope this paper can be published, after the little issues that I flagged are corrected.

Reviewer #2 (Remarks to the Author):

This manuscript attempts to establish the savings in flood protection afforded by coral reefs. The study is another attempt to ascribe monetary value to one of the ecosystem services provided by coral reefs to highlight their importance. Similar attempts have been attempted for more than a decade including attempts by the World Bank in 2005. The study also attempts to provide a global overview of flood protection savings. While the aim is laudable I believe the results are not novel. I have serious reservations concerning the gross assumptions made in the study and flaws in logic of reef systems that these expose.

These results are novel; there has been no other global, process-based assessment of the flood protection services of any marine or freshwater ecosystem. We have been supported by the World Bank for the past 3 years specifically to review what has been done before and to identify how to do a rigorous valuation relevant to nations.

1. The assumption made is that globally all reefs will lose 1m of reef due to storms and other impacts. There are several concerns related to this assumption:

We appreciate that we needed to make it much clearer why a ‘without reefs’ scenario was needed for the assessment and why a 1m change was not only reasonable but also conservative relative to the other site- and indicator-based assessments of coastal protection services. Further we have added more references that show ongoing, current declines in reef cover and height across all reef environments.

Line 286- “Estimating reef benefits. To examine the current value of reefs for coastal protection, we compared flooding under current conditions, “with reef”, to the flooding in a scenario “without reefs”. In our “without reefs” scenario, we do not assume the loss of the entire reef habitat; we assume only the loss of the top 1m in height across the reef bathymetric profile. Many ecosystem service assessments assume the entire loss of a habitat for estimating benefits. For example, the replacement cost method, which is the most commonly used method for estimating the benefits from mangrove and reef habitats¹², identifies the flood reduction benefits from habitats by estimating the cost of replacing them with seawalls or breakwaters. Many problems have been identified with the replacement method and it provides estimates of values 10 times higher than the recommended Expected Damage Function approach that we follow^{12,40}.

The “without reefs” scenario is not meant to be a prediction of site-specific trajectories for reefs, but nonetheless this level of loss is already observed to be happening^{14,25,26} and is conservative relative to future predictions of reef loss²⁸⁻³⁰. In addition to the widely observed declines in coral cover, growth and condition, all of which affect reef height^{20,21}, new measures of seafloor elevation show that bioerosion and carbonate dissolution are degrading height across all reef habitats including on reef flats²⁶. Damage from storm events can also create losses in reef height of 1-3 m^{75,76} and can devastate whole shallow reef frameworks⁷⁷. Past storms have removed many branching and massive corals at the shallowest depths^{24,75}. Shallow corals have evolved with intermittent storms and can recover from them, but this is more difficult when reefs are exposed to multiple stressors²⁴.”

a. There is a paucity of literature to back up this assumption. Alvarez et al. (2009) is cited as the main reference to support such changes. This study was limited to the Caribbean only and examined different reef locations. Such conclusions cannot and should not be transferred to other reef systems globally, which has responded very differently to Caribbean reefs over the past century. Indeed, there are many more studies that have shown that reefs have grown in the past century – why not use those scenarios? We have added more references that show ongoing, current declines in reef height across all reef environments- though again an ecosystem service assessment is not about current trajectories only a

reasonable “without reefs” scenario. We agree that more reef height measures should be taken and this is a clear point in the Discussion and from our sensitivity analyses.

The evidence is clear that there has been widespread reef degradation and that this is leading to loss of corals (particularly shallow corals) and reef height. As noted above, we have provided more references in this regard. Every study that we are aware of that examines coral reefs across regions, oceans and globally indicates a very clear pattern of declines in living coral cover, and where measured, declines in height. There has been some site-specific recovery from impacts such as bleaching from climate change (and site specific reef growth), but there is very broad agreement in the literature on the general decline of coral reefs.

b. The assumption above also assumes that all reefs have a 1m thick veneer of living coral. This assumption is badly flawed. The manuscript does indicate that removal of the reef framework is not considered in the model. This admission undermines the contention that 1m of reef can be removed from all reefs globally. For example, the majority of Pacific reef systems have only a very thin veneer of living coral (<0.3m). Furthermore, most Pacific and open ocean reef systems do not have coral on the reef rim. Rather they are colonised by coralline algae (which can grow higher in elevation than coral) due to the high nature of the reef rims.

At the beginning of the paper, we have added references to indicate that both living corals and the underlying substrate are losing height as follows;

Line 41- “Reefs have experienced significant losses globally in living corals and reef structures from coastal development; sand and coral mining; overfishing and destructive (e.g., dynamite) fishing; storms; and climate-related bleaching events^{8,20-23}. There is clear evidence of reef flattening globally from the loss of corals and from the bioerosion and dissolution of the underlying reef carbonate structures^{14,24-27}. Not all reefs are declining, and reefs can recover from bleaching, overfishing and storm impacts, but the overall pattern of significant losses across geographies is clear^{20,21}. Scientists and international agencies, including the Intergovernmental Panel on Climate Change and the World Bank, have expressed grave concern about the current and future condition of coral reefs, and the loss of the benefits they provide²⁸⁻³⁰.”

We agree that there should be more field measures, but it is also not surprising that limestone erodes in high energy, increasingly acidic environments with many active bioeroding animals (e.g., burrowing bivalves, crustaceans and worms). However, even we were surprised by the high rate of substrate loss indicated by the new data.

We account for the present status of the wave breaking function of reefs in their current extent and depth with the best global distribution and bathymetry data available. The reviewer is correct that some Indo-Pacific reefs do have flats with less living emergent structure than observed in the Caribbean. But the rest of these shallow reefs down to 5m or more,

have extensive emergent corals in the Indo-Pacific. Further we show from a new study that even reef flats are losing significant height from bioerosion and carbonate dissolution.

c. The critical zone for wave breaking and reduction of wave energy is the upper reef crest and reef flat. The literature cited regarding reef development generally does not refer to these zones. Indo-Pacific reefs are generally depauperate in coral on these surfaces (or have an extremely patchy and thin veneer). Consequently, removal of living coral is unlikely to change the elevation of the reef crest and reef flats substantially in these zones. Given the Indo-Pacific accounts for the vast majority of reef globally, the assumption of the study is invalid. The loss of 1m of coral from the forereef slope will also have minimal impact on wave transformation processes.

There is clear evidence that storms globally damage and remove massive corals including in the Indo-Pacific, which are cited in the paper (e.g., Harmelin-Vivien 1994, De'ath et al. 2012, Puotinen et al. 2016). This indicates that reefs and reef height are regularly being impacted in areas critical for wave breaking and flood reduction (because wave energy is removing large corals).

While the authors may argue the actual reef responses are irrelevant to the point being made that reefs provide flood protection I disagree. It is irresponsible to set up an academic argument founded on false assumptions. In general the manuscript lacks convincing insights into reef development and change, or reef hydrodynamics and without such a foundation the economic analyses really have no value.

As noted above (i) we have provided literature to back up the fact that corals are losing height across environments, (ii) we discuss that ecosystem service assessments require estimates of a "without habitats" scenario and ours is more conservative than other published estimates, and (iii) the literature on the overall pattern of major declines in reef condition and height is clear globally. As indicated above the identification of alternative scenarios is required for valuations and is not a prediction of site-specific trajectories. Nonetheless we have identified a very reasonable and conservative 'without reefs' scenario and one that is unfortunately playing out for many shallow coral reefs globally. Further we have provided the sensitivity analyses around bathymetry to help readers evaluate likely differences in other scenarios (if they have other alternatives in mind) and to make the point that bathymetry is critical for all global coastal flooding models (and we have done a better job of including it than any other global flooding model).

2. The hydrodynamic analysis is also simplistic and based on limited studies. What is not reflected in the study is that substantial changes in water depth across reefs fundamentally changes the types of waves and wave frequencies impacting shorelines. These changes radically alter the types of waves and nature of flooding of coastal lowlands. The manuscript fails to recognise these critical differences. The manuscript uses outdated coastal engineering tools to assess the flooding.

As described above and further here, we provide the most rigorous global coastal flooding analysis available. And in fact, ours is the only global analysis that includes reef bathymetry – i.e., we explicitly recognize that water depth changes across the reef. We have provided greater clarity around these issues in text (including limitations).

Line 152- “Our coastal flooding have analyses several significant improvements over other recent global flooding analyses^{2,5,6} including downscaling to a 90m resolution; consideration of hydraulic connectivity in the flooding of land; the use of 30 years of wave, surge, tide and sea level data; reconstruction of the flooding height time series and associated flood return periods; inclusion of nearshore bathymetry and reef profiles; and the use of country-specific adjustments to allocate GDP per person. Major remaining constraints for all global coastal flooding models include the consideration of flooding as a one-dimensional process and the difficulty in representing flooding well in smaller islands. Since reefs are offshore structures that often run parallel to shore, a one-dimensional flooding approach (i.e., where water moves only perpendicular to shorelines) is a reasonable assumption for a global model of reefs. These approaches do not consider nonlinear effects such as energy transfer among different wave frequencies that may contribute to changes in flooding. However, the analysis of these effects is only possible at local scales (e.g., bays) using phase resolving models that require high resolution bathymetry and large computational efforts^{12,15,49,50}. This approach is not feasible at the global scale because of computational capacity and the lack of high resolution bathymetric data and especially if risk is to be evaluated in terms of annual expected damages. Although we used a finer resolution grid size than prior global flooding analyses, some of the smallest island nations are not well sampled.”

We are fully aware of the hydrodynamic and morphodynamic processes behind the interaction of waves and coral reefs including the most recent outcomes from local field experiments and numerical simulations. However our analysis; 1) is at the global scale; 2) requires the integration of very diverse data sets; and 3) is based on a probabilistic approach requiring the analysis of long-term time series to calculate return periods.

The reviewer suggests that that our modelling approach does not account for changes in the “type of waves and wave frequencies”. Most of these processes are dominated by nonlinear effects. It is well-known that these effects can only be solved properly by using wave phase resolving models. Phase resolving models have the advantage that they are able to solve the wave shape and consequently require to be solved using a spatial high discretization to provide enough detail of the wave shape. That means that independently of being based on potential flow based models or advanced RANS-type models, bathymetric high resolution commensurate with the model discretization and longer computational demand is needed. However, 1) high quality bathymetric resolution is not available globally and 2) the number of sea states that can be run along 70,000 km using these kind of models, is statistically insignificant to obtain extreme distributions. Moreover, these models are not superior to our phase-average model when simulating the damping induced by the living coral. In this kind of model it is also purely based on a friction, Chezy or drag coefficient approximation. In our calculation we take into

account the effect of both the short wave and long wave components in the runup formulation. Consequently, the hydrodynamic modelling approach has to be selected finding the balance between a process-based model able to reproduce the most relevant processes consistent with the quality of the available data sets a global scale and needs to be computationally efficient to consider 70,000 km and 32 years long time series to obtain probabilistic information. In sum, to our knowledge we have run the most rigorous global coastal flood model to date because:

- An estimation of the annual expected damages requires the calculation of the full flooding height distribution across different storms return periods. Thus, we developed the required long time series of flooding heights along a 70,000 km shoreline.*
- To our knowledge this is the first time ever in which a flooding analysis at global scale is based on a 32-years combination of time series of waves, storm surge and astronomical tide at high resolution. It is also the first time ever that a 32-years time series of flooding heights is reconstructed at the shoreline providing a statistical significance to the results superior to previous studies relying on other much simpler approaches.*
- To our knowledge it is also the first time ever that a global scale bathymetry has been modified to include the effect of coral reefs.*

In the new version of the manuscript we have been clear about the assumptions and limitations.

3. The manuscript also assumes that the coastal margin abutting reef systems is inert and will simply be drowned with increasing water level. In essence elevated water levels simply flood the coast to a different bathymetric contour line in the often used ‘bathtub’ approach. Such an approach has been widely criticised as it fails to reflect the dynamic nature of coastal margins. Indeed, numerous studies have shown that coastal margins are dynamic and can change their elevation with respect to high runup levels.

We have indicated more clearly that we did not follow a bathtub approach in the Discussion and Methods.

Line 152- “Our coastal flooding have analyses several significant improvements over other recent global flooding analyses^{2,5,6} including downscaling to a 90m resolution; consideration of hydraulic connectivity in the flooding of land; the use of 30 years of wave, surge, tide and sea level data; reconstruction of the flooding height time series and associated flood return periods; inclusion of nearshore bathymetry and reef profiles; and the use of country-specific adjustments to allocate GDP per person.

Line 313 **“Calculating land, people and built capital flooded.** We assessed flood heights along each coastal profile and then identified the area flooded within each coastal study unit. We extended the flood heights inland by ensuring hydraulic connectivity between points at a 90m resolution; a significant advance over more common bathtub approaches in earlier global flooding models.”

With regards to the point that we do not account for “the dynamic nature of coastal margins”. This is correct but as before none of the global models account for this incredibly complex effect, because it is simply not possible given existing data and computational capacity.

Collectively, the detailed analysis stacks up multiple gross and incorrect assumptions that render the analysis of flooding meaningless. Consequently, the results and discussion sections are full with generic and over-extended assertions of the magnitude of coastal flooding, likely impacts and their future changes.

Our paper and analyses should be reviewed (i) relative to other large scale flooding models published in high-profile journals and (ii) the available data and computing capacity. We have published the best and most rigorous global assessment of these flood protection benefits.

Currently, the desire to make global assertions of impacts is simply not believable on the basis of the assumptions and evidence presented. I would have thought a better approach may have been to use more detailed analysis of selected sites where losses have been documented. Application of latest numerical modelling approaches could then highlight precisely how flood wave processes will change and influence coastal hinterland. Similarly, this would have allowed the article to explore what savings are made at sites where the reef has been accreting over the past century.

We believe in the value of site-based studies for making site-based assessments, and we have used these numerical models at sites before. However, it is not possible to make global or even national-level assessments of flood risk and protection benefits from a few study sites. Site based studies cannot be upscaled because of the variability in the conditions - many of which we capture in our global model such as the heterogeneity of ocean climate conditions, bottom morphologies, coral reefs, shoreline morphology, topography, population distribution and assets distribution. We strongly believe that despite our simplifications, our approach provides the most robust global analysis to date and that it is, in fact, better than similar global analysis of flooding impacts on population and assets recently published in this or similar journals.

On the basis of the above comments I believe the manuscript should be rejected at this time.

Reviewer #3 (Remarks to the Author):

The manuscript submitted by Beck et al. is a really interesting and important contribution. They demonstrate the risk reduction benefits of coral ecosystems at a global scale, using process-based methods. They also provide outputs in units that will resonate with different readers or stakeholders. The information provided will be useful to coastal managers, government officials, engineers and scientists alike. However, although the authors seemed to be very careful in their analysis, I found the paper hard to follow at times, both in its style, but also because of lack of clarity on in many of the paper's sections.

We agree and we have made many changes to improve clarity.

I suggest that during their revision, the authors spend time improving the overall style and readability of the manuscript. Special attention should be paid to the use of words, transitions between sentences and paragraph, paragraph structure, and overall flow of information. For example, in the paragraph starting in Line 95, where the first sentence talks about future sea level rise, the second refers reef elevation loss under current conditions, and the third refers to sea level rise. This strange arrangement led to some confusion in my reading of the manuscript, and so can potentially confuse other readers.

We agree and have made improvements throughout including the specific one suggested above.

Another source of confusion is the loose uses of the words climate change and flooding. For example, in the Abstract, the manuscript reads “Climate change will increase flood risk [...]” (Line 24). This sentence is not clearly linked to the previous one, which was discussing countries with most to gain from reef conservation. But it is also unclear why the authors say that climate change will increase flood risk: is it because of increased rainfall, higher waves, stronger hurricanes, sea-level rise? Similarly, in the first paragraph, sentences 2 and 3 (Line 34) don’t really follow each other, and once again the term “climate change” is used loosely. It is left to the reader to guess which climate change impact will increase flooding. The same observations are valid for the use of the word “flooding”. To most readers, “flooding” usually means that water elevation at a particular location is higher than usual, and usually it is caused by intense rain, or by a storm surge. Here, most of the flooding that is reduced by corals is due to a reduction in wave runup, which is one portion, and generally not the most important, of flooding. So I suggest that authors pay attention to readability of their manuscript, and take the time to define as early as possible how different terms are used in the manuscript, such as flooding or climate change.

We have been clearer in the abstract, results and discussion that the climate change impact that we examine is sea level rise. We have been clearer up front about the main effect of coral reefs is in wave breaking.

As noted below we do consider storm surge in the calculation of flooding and the propagation of waves over coral reefs. We have been clear throughout that reefs themselves primarily reduce flooding through their effects on run-up. Indeed on coral reef coastlines wave run up is a major driver of flood risk.

Line 266- “Flood height, flooding levels and return periods. Flood height onshore is calculated from the combined action of mean sea level (MSL), astronomical tide (AT), storm surge (SS), and wave run-up (Ru).”

Other stylistic comments, which are not exhaustive:

1. Please revise the use of the word “underwater” in Line 64. That sentence is not clear. *Done.*
2. Are there “a couple” (i.e., two) national studies (Line 64) to quantify ecosystem benefits or “at least a couple”? *We indicated “a couple” because we are only aware of two that can be identified as at*

the national scale.

3. Expand on what “other impacts” are causing loss of coral structure (Line 80). Are human activities part of these causes? This might be important for intended audience to know. *Done.*

4. Replace “In” by “For” when describing results for 25 and 100 year events (Lines 90 and 92). *Done.*

5. Check use of the word “currently” in line 90. *We have removed “currently”.*

6. On Line 137, instead of “reduce wave energy dissipation”, why not say “increase transmitted wave energy” so it can be understood by wider audience? *We have improved this sentence.*

7. On line 142, the expression “reducing flooded areas” is confusing. Please be more clear. *We have removed.*

8. What are Small Island Developing States (Line 147-148)? Are they the same as the nations discussed earlier (Line 119)? Please clarify. *We have removed and replaced with “many small island and developing States”.*

9. The word “business as usual” is usually used in climate projection. Maybe use another word? (Line 152). *Most of the future projections about reef loss that we cite cover primarily climate-related warming and acidification; we think the term is appropriate in this context.*

10. Provide a reference for the claim that 1 m of reef height can happen in the near term (Line 154). Also, this sentence give a sense of urgency that seems out of place. Should it be in the motivation section of the introduction? *As noted above we have made many changes to indicate that these losses have been observed, but that more importantly we use the 1m change in bathymetry as a modest “without reefs” scenario to estimate the value of reefs.*

11. The last two paragraphs of the Discussion seem out of place. Please review for readability. *We have substantially revised the discussion. We believe that some of the key points in the last paragraphs were important to preserve to provide context for the more general readers of Nature Communications on why the work matters.*

In addition to these general comments, I suggest that the authors review their manuscript and pay attention to how they present information. I feel that the Introduction does not properly motivate the question being researched, and the discussion section feels unstructured in general. I suggest that the author revise the order of their arguments and the points and connections they want to make. Other technical comments below:

We agree and have extensively revised the introduction.

1. The introduction does not clearly explain why it is important to look at how much coral reefs reduce flooding caused by wave runup. The authors mention coral reefs in the first paragraph of the introduction (Line 36), then again later in paragraph before last. However, it is not revealed until the last sentence why it is important to look at the importance of reefs in risk reduction, which is that reefs, globally, are losing rugosity, elevation and will be more at risk of degradation as time passes. I suggest that the authors motivate this study as early as possible, probably in the first or second paragraph *We agree and have extensively revised the introduction to make these points more clearly.*

2. It is unclear how modeling scenarios were created. They are described in the last paragraph with very little information to justify them. Information about the flattening of reefs is provided after the scenarios are presented, and the reference for the flattening is only for the Caribbean. What about the other regions of the world? Also, in that last paragraph of the introduction, the authors mention a 1-meter decline in reef height, but this value does not seem to be justified, and it is unclear if it applies in the future or in the present. I suggest that the authors spend more time carefully explain the rationale behind their scenario, then explain clearly what these scenarios are.

We agree and as noted above we have extensively revised to make the case in multiple places that (a) to estimate benefits a “without reefs” scenario is required, (b) that we have chosen a reasonable scenario, (c) that these kind of losses have been observed and are predicted to worsen and (d) that we have provided sensitivity analyses of bathymetry for the reader to assess how changes in bathymetry will affect results.

As one example we have extensively revised the Methods as follows;

Line 286- “Estimating reef benefits. To examine the current value of reefs for coastal protection, we compared flooding under current conditions, “with reef”, to the flooding in a scenario “without reefs”. In our “without reefs” scenario, we do not assume the loss of the entire reef habitat; we assume only the loss of the top 1m in height across the reef bathymetric profile. Many ecosystem service assessments assume the entire loss of a habitat for estimating benefits. For example, the replacement cost method, which is the most commonly used method for estimating the benefits from mangrove and reef habitats¹², identifies the flood reduction benefits from habitats by estimating the cost of replacing them with seawalls or breakwaters. Many problems have been identified with the replacement method and it provides estimates of values 10 times higher than the recommended Expected Damage Function approach that we follow^{12,40}.

The “without reefs” scenario is not meant to be a prediction of site-specific trajectories for reefs, but nonetheless this level of loss is already observed to be happening^{14,25,26} and is conservative relative to future predictions of reef loss²⁸⁻³⁰. In addition to the widely observed declines in coral cover, growth and condition, all of which affect reef height^{20,21}, new measures of seafloor elevation show that bioerosion and carbonate dissolution are degrading height across all reef habitats including on reef flats²⁶. Damage from storm events can also create losses in reef height of 1-3 m^{75,76} and can devastate whole shallow reef frameworks⁷⁷. Past storms have removed many branching and massive corals at the shallowest

depths^{24,75}. Shallow corals have evolved with intermittent storms and can recover from them, but this is more difficult when reefs are exposed to multiple stressors²⁴.”

3. Does the reef distance to shore matter? The authors did not provide any information about the types of reefs that provided more protection than others, and why. This might be interesting for those trying to understand what matters most in reef restoration.

Distance to shore is already accounted for in the model as we use the spatial information that is available on reef location and bathymetry. We have made the importance of characteristics such as reef height and rugosity clearer.

4. In sentence starting on Line 110, the authors claim that “hotspots areas” suffer many storms. Can they comment on the ability of the reefs to withstand these storms and recover in between storms so they can deliver the same level of protection through time?

We have provided more information and a reference on this topic as follows;

Line 300- “Damage from storm events can also create losses in reef height of 1-3 m^{75,76} and can devastate whole shallow reef frameworks⁷⁷. Past storms have removed many branching and massive corals at the shallowest depths^{24,75}. Shallow corals have evolved with intermittent storms and can recover from them, but this is more difficult when reefs are exposed to multiple stressors²⁴.”

5. I mentioned this above, but it is important that authors clearly explain how reefs reduce flood levels, namely, that they reduce wave runup, in order to clear set expectations in readers. The first sentence of the discussion (Line 128) could be interpreted as reefs reduce storm surge levels to fast readers.

As indicated above, we are much clearer on the effects of corals on wave run-up throughout the paper including in the introduction.

6. The authors discuss stressors to corals reefs at the beginning of the Discussion section (Line 133), then again in the third paragraph of that section (Line 146), then discuss other ecosystem services on line 179. Maybe move this type information to the introduction to help motivate the manuscript, or at the end of the Discussion section?

We agree and have moved more of this information in to the Introduction and clarified in the Discussion.

7. On Line 137, the authors mention that impacts on reefs can leave smooth limestone structures. It is my understanding that those consequences can be observed decades after impacts started. If that’s the case, this statement adds to my confusion as to the motivation and interpretation of the scenarios: are we looking at current conditions, future conditions with 1 m sea-level rise and loss of rugosity, or future conditions with loss of rugosity without sea-level rise? My apparent confusion argues for more clarity in the introduction on what is exactly measured and why.

We have been much clearer about the scenarios and motivation as indicated above.

8. How is the 1 m height decrease exactly modeled? The text suggests that bathymetry is lowered by 1 m (Line 77), but Figure 1 , Panel B, seems to suggest that the bathymetry remains unchanged, but the friction coefficient is reduced. Please clarify and be consistent throughout the text.

We have clarified in the Methods that our “Without Reef” scenarios include 1m reductions in height across the bathymetric profile as well as reductions to the friction coefficient. Figure 1B shows declines in reef height across the profile; we are not certain there is a better way to show this change graphically (and at any rate we are clearer in the text throughout).

9. On Lines 138-140, the authors mention, once again, that corals are flattening in the Caribbean, but they use the same reference (ref. 38) to mention that reefs are declining in height in other parts of the world. I suggest that the authors complete a more exhaustive literature survey on the subject and find references for regions in other parts of the world, or focus solely on the Caribbean.

We have provided many new references about observed declines in shallow water corals indicating declines, damages and losses of these corals and coral reef height globally.

10. The authors mention that their estimates are conservative, on Line 152. Please explain, because the model has many sources of uncertainty (water depth, water elevation, friction coefficients, etc.)

We have more clearly indicated why they are conservative particularly with regards to other valuations of ecosystem services which assume the entire loss of habitats (e.g., replacement cost method, which is the most commonly used valuation approach).

11. The discussion section does not address sufficiently the limitations of the study, and the how to interpret results properly. The methods, although sound, are not precise enough to correctly capture all the nearshore processes that affect inundation, so results are a 1st estimate, and cannot be used to make important decision. Since the authors recommend that nations use their results in their general accounting, it is important to be clear on the limitations of these results.

We have clarified that we have significantly improved on other recent hi-profile global coastal flooding analyses and indicated some of the key limitations in the Discussion.

Line 152- “Our coastal flooding have analyses several significant improvements over other recent global flooding analyses^{2,5,6} including downscaling to a 90m resolution; consideration of hydraulic connectivity in the flooding of land; the use of 30 years of wave, surge, tide and sea level data; reconstruction of the flooding height time series and associated flood return periods; inclusion of nearshore bathymetry and reef profiles; and the use of country-specific adjustments to allocate GDP per person. Major remaining constraints for all global coastal flooding models include the consideration of flooding as a one-dimensional process and the difficulty in representing flooding well in smaller islands. Since reefs are offshore structures that often run parallel to shore, a one-dimensional flooding approach (i.e., where water moves only perpendicular to shorelines) is a reasonable assumption for a global model of reefs. These approaches do not consider nonlinear effects such as energy transfer among different wave frequencies that may contribute to changes in flooding. However, the analysis of these effects is only possible at local scales (e.g., bays) using phase resolving models that require high resolution bathymetry and large computational efforts^{12,15,49,50}. This approach is not feasible at the global scale because of computational

capacity and the lack of high resolution bathymetric data and especially if risk is to be evaluated in terms of annual expected damages. Although we used a finer resolution grid size than prior global flooding analyses, some of the smallest island nations are not well sampled.”

12. In the Nearshore Hydrodynamics Methods section (Line 244) please clarify that this is a 1-D model, and clearly state its limitations. Also provide some information about the formulation of D_f .

We have clarified that this is a 1-D model and we have clearly indicated that the parameter estimates for D_f come from Sheppard et al. 2005.

13. The authors do not provide information on the accuracy of their topography data (Line 255-260). This might be important to put results into perspective, especially in light of the results of their sensitivity analysis.

In text we have provided further clarification as follows;

Line 258 – “We combined topographic⁶⁸ and bathymetric data into an integrated set at each geography. We use the shuttle radar topography mission SRTM 90m database for global elevation⁶⁸, which has been identified as the best globally available digital elevation model and has been used in other regional and global flood models^{2,5,6}.”

The SRTM provides elevation for every 90m 90m cell and is the highest resolution dataset we use. These datasets have been used to assess people in Low Elevation Coastal Zones, e.g. (Neumann et al., 2015) and in other flood risk assessments (e.g. Hinkel et al., 2014). Hinkel et al. (2014) and Reguero et al., (2015b) identify the SRTM 90 data to be the best available.*

14. The authors do not provide sufficient information on the modeling of storm surge (Line 262) in their estimation of flood elevation. This is important as storm surge is an important component of flooding. Even in the SI, the authors do not explain how they treat the storm surge component of their dataset. The references that they provide are all for waves (SI Lines 170). This comment again argues for more
We are now clear in the very beginning of the mss that reefs primarily act to reduce flooding through wave breaking and friction. As noted at the beginning of the Methods, our data on global waves and water levels does account for surge in the water level estimate and this is carried through to estimates of wave propagation and land flooding. While we do not account for the transformation of the storm surge by the coral reefs, we do include the effect of storm surge on wave transformation over the reefs and consequently on the flooding levels. On coral reef coastlines wave run up is a main driver of flood risk, and wave run up is the main driver affected by reefs.

We have added another reference on the storm surge to clarify where we get the storm surge data, which is the DAC global surge dataset.

15. The runup expression derived by Stockdon is only valid for sandy beaches (Line 265-266), on relatively exposed coasts. It contains an estimate of swash and setup generated by infragravity and

incident waves. In their manuscript the authors assume, without justification, that reefs affect both incident and infragravity wave processes, and that all coasts are sandy, exposed, beaches. Also, the authors use the nearshore slope (Line 268), whereas the equation asks for the foreshore slope, which might lead to large errors, especially given the general lack of accuracy of nearshore depths in global bathymetry dataset. This is a stretch of the runup formulation provided by Stockdon, and the authors do not provide sufficient information on the computation of runup, and do not justify the limitations of their approach.

We agree that further clarification is needed and have clarified as follows.

Line 266- “Flood height, flooding levels and return periods. Flood height onshore is calculated from the combined action of mean sea level (MSL), astronomical tide (AT), storm surge (SS), and wave run-up (Ru). In our model, coral reefs affect flood height by modifying the wave properties and hence the run-up. We estimated the wave run-up⁷² as:

$$Ru_{2\%} = 1.1 \cdot (0.35m\sqrt{H_0L_0}) + 0.55 \cdot (H_0L_0\sqrt{0.004 + 0.563m^2})$$

where $Ru_{2\%}$ is the wave run-up exceeded on average only 2% of the time for a given sea state, H_0 is the significant wave height offshore, L_0 represents the deep water wave length, and m is the bathymetry slope from the shore to the foreshore breaking point. In comparisons of this equation relative to numerical models, Stockdon and others⁷³ show that wave set-up was accurately predicted by the equation. There is no available formulation for the swash component (i.e., the second term in the Stockdon equation) in reef environments. To estimate H_0 and m we first identified the breaking point depth (h_b) at which a given wave breaks offshore on the reef (with height H_b) and then assumed a beach slope (m) from h_b to the shore (0m elevation). We then identified H_0 as the wave height that would break at h_b on the beach with slope m . The relevant processes that generate the swash component occur in the breaking process and the surf zone; our approach assumes that the processes from the reef breaking point will behave similar to those on an equivalent beach slope with similar breaking point.”

In recent publications Plant and Stockdon (2015) and Stockdon et al (2014) evaluate the application of the run-up equation under different conditions and compared it against more advanced numerical models and observations. One of the main conclusions is that “if it is not known a priori which unresolved processes are important, predictions are likely most robust if they fit a wide range of conditions as was done in Stockdon et al (2006).” Moreover, they claim that by using the full model including (1) wave-driven setup and swash motions due to both (2) incident-band and (3) infragravity-band frequencies, run-up on either very flat or very steep beaches could be predicted using the equation at either end of these kind of beaches by neglecting certain processes. Furthermore, they recognize “Discrepancies between parameterized and observed runup can be attributed to a number of factors, including uncertainty in nearshore wave height and uncertainty in beach slope and beach topography, which should be measured synchronously with overwash events-but rarely are”. Consequently, in the absence of high quality beach topography we believe that our estimation of m is the most

consistent with the approach followed by Stockton et al. in the derivation of their predictive formulation.

In Stockdon et al. (2014) they compare the approach in the run-up equation with Xbeach (a numerical model) and they showed that while some effects could only be discerned with high res models and data that wave setup was accurately predicted by both the parameterized equation and numerical simulations. Consequently, we believe that by applying Stockdon et al (2006) we are using the best available prediction formula able to capture a very broad range of conditions and with computational and data requirement affordable at global scale.

Stockdon, H.F., Thompson, D.M., Plant, N.G., Long, J.W. (2014). Evaluation of wave runup predictions from numerical and parametric models. Coastal Engineering 92, 1-11.

Plant, N.G., Stockton, H.F. (2015). How well can runup be predicted? Coastal Engineering 102 (2015), 44-48.

16. In their application of the runup equation, the authors do not properly explain how they estimate the equivalent deepwater wave (Line 267). I suggest improving the content of the manuscript using some of the language used in the SI (SI Line 116)

We agree and have clarified as above (see point 15).

17. In their application of the runup equation, the authors seem to assume that there is always a wave setup generated at the coast. However, studies by Lowe et al (2010) suggest that setup, and thus runup, shoreward of reefs is a complex function of reef geometry. Please clarify the limitations of the existing study.

Our 1-D approach does assume that there is always a wave set up at the coast and as noted above (reviewer 3, point 11) we now specifically discuss the limitations in this approach in the Discussion.

We recognize that in cases like Lowe et al (2010) where analyses are done at the scale of an individual reef and with high res bathymetry that it is possible to assess complex interactions with numerical models. Neither these data nor models are possible at global scales.

18. I found the explanation of the choice of different friction coefficients for reefs, and their use in the different scenarios lacking (Lines 288-295, SI Lines 150-154). In particular, no justification was given for the choice of $C_f=0.08$ for reefs without living corals (Line 295, no mention in the SI). As mentioned earlier, I suggest that the authors clarify their methods how they treat the different scenarios.

We have more clearly indicated that these values are from Sheppard et al. 2005 and we have included these values directly from that paper in Table S3. The value of 0.08 is the friction for a sand bottom, that we assume for "no reef" cover (as in Sheppard et al 2005).

19. I appreciate the authors did a sensitivity analysis. I suggest including nearshore slope in their parameters spaces, given that they did not apply the runup equation correctly. I also suggest that they use consistent language in the SI to compare results of their analysis (SI Lines 221-228).

As noted above we have explained much better how we have identified the key parameters in the run-up equation. While our sensitivity analyses do not include multiple slopes they do consider two different key reef geometries with different slopes from fringing and barrier reef geometries and in general we do not find great differences in the results for these two geometries across the three key parameters we have assessed. As noted above, we also cite Stockdon and others, who examine many of the sources of uncertainty in the widely applied run-up equation.

20. The horizontal axis or text of Figure S3 is difficult to understand. I suggest the authors improve it. *We have removed the text at the very bottom of Figure S3 as we agree that it did not help in clarification.*

21. The regions north of New Zealand do not seem to be included in the analysis (Figure S4). Can you clarify why?

We are not sure what regions the reviewer refers to. All areas are included that have coral reefs (as identified from the best available global database) and populations (as identified in the best available population database).

Reviewers' comments:

Reviewer #1 (Remarks to the Author):

Thanks to the authors for this improved version. It mostly answers my concerns and I think the paper could be published as it is.

I have a couple of small suggestions:

Page 5, line 83 "reefs provide more benefits for lower intensity frequent storms": I would add "in relative terms" since the absolute value increases in Figure 3. Also, it may be useful to add a panel to Figure 3 with the effect in percentage point, to show the decrease mentioned in the text.

At the end of the introduction, the authors may want to add a couple of sentence recognizing that a global assessment cannot get to the level of detail that a local analysis can afford. This may help answer the concerns of the second referee.

As a side note, I've been in the authors' shoes a few times, with reviewers doing local and high-resolution analysis expressing concerns about a global analysis that by construction cannot get to the same level of details. While I am not a specialist of the issues at stake in the comments of the second reviewer, I would like to support the authors. While local high-resolution studies are critical, they do not replace global assessments based on simpler methodologies. We do need a hierarchy of studies, from global to local scales, and from simple to complicated models.

While the global studies will have to make simplifications (for reasons related to data availability to computation requirements), they can add a lot of value in terms of research, and in practical terms they are very valuable to identify hot spots and quantify the issues. And research is iterative: the authors or other scholars (maybe the second reviewer?) may be able to start from the proposed study and improve it by adding more mechanisms and better data.

While reviewers cannot ask global studies to be as detailed as local one, they must ask authors to be very clear on the assumptions and simplifications. On this, I feel that the authors are doing a great job, and their study is very transparent.

I think the results of the sensitivity analysis should be better communicated. In particular, I could not find the impact of various assumptions on the economic assessments. It would bring a lot of confidence in the results if the authors could show that different assumptions do not completely change their results. I'm not sure how the changes in run-up would translate into changes in economic losses due to coastal floods.

And it would be useful to include other uncertainties – for instance the assumption that 50% of the value of a building is lost when flooded by a 1-meter flood is probably acceptable for many buildings in developing countries, it may not be the case for modern buildings (especially because many buildings in places that are regularly flooded have stilts or do not fully use the ground floor to minimize losses).

The value of 50% may be even more overestimated since the “flood” is here defined by the wave height – not as the level of permanent flooding. So the authors' assumptions is different from most flood vulnerability curves, which use the permanent water level as an input (not the wave height). I would suggest to the authors to stress test their results to this assumption.

Reviewer #2 (Remarks to the Author):

Thank you for forwarding the revised manuscript for my reconsideration.

To answer your question directly I do not believe the amendments have adequately responded to my concerns in the revision. I highlight several specific elements in this regard.

1. Loss of reef elevation. The authors state that globally there is widespread consensus that reefs are degrading. It is true that the weight of published material does suggest that the proportion of living coral is decreasing. The authors further assert that coral degradation and structural reef loss are one in the same. This is not the case. The authors have increased the number of references to support this statement. However, they do not all measure reef loss. The Alvarez article certainly does. However, the Perry and Morgan article simply infers this might be an outcome but present no quantitative data to support this. Only include articles that actually measure structural loss. Many articles suggest this might be an outcome but fail to demonstrate this.

2. I raised a specific concern related to the nature of wave breaking with reefs and the fact that across much of the Indo-Pacific the zones of critical wave breaking is depauperate of coral (comment 1c). The authors respond with:

“There is clear evidence that storms globally damage and remove massive corals including in the Indo-Pacific, which are cited in the paper). This indicates that reefs and reef height are regularly being impacted in areas critical for wave breaking and flood reduction (because wave energy is removing large corals).”

This response represents a complete lack in understanding of the nuances between individual coral growth and reef platform development and change, which is at the heart of the scenarios developed. Simply because a storm may pluck corals from the forereef does not equate to structural loss of reef surfaces. Indeed in Cyclone Bebe, extreme waves removed more than 1 million cubic metres of coral and rubble from the forereef of Funafuti atoll. This material was spread across the reef platform surface. The net effect was to increase the reef surface by >0.5 m due to the deposition of rubble as a blanket across the reef surface.

3. The authors claim the model is not of the ‘bathtub’ variety. However, I can find no element in the approach that allows for the landform feedbacks in the system that would potentially offset future flooding.

4. Hydrodynamic modelling. The authors appear to agree with my concerns related to hydrodynamic modelling. They state that the necessary level of analysis to fully understand wave reef interactions "...is only possible at local scales (e.g. bays) using phase resolving models that require high resolution bathymetry and large computational efforts."

The argument being put forward seems to rest on the global scale of the model, which the authors acknowledge is at too coarse a scale to represent wave-reef-coast interactions very well. Furthermore, the authors appear to argue that this model approach is an improvement on other approaches.

I disagree that the paper should be evaluated relative to other large scale flooding models, as asserted by the authors. The substance of the article will have significant international traction. The reef community deserves outcomes that can be relied upon at the reef scale. The admission that the model cannot yield this - would seem to be a fatal flaw.

I fully understand the attractiveness of attempting to value what coral reefs can afford, from multiple angles. However, I am unconvinced that a better, but wrong answer (which the authors acknowledge, due to the computational complexities), is something that should be released to the broader science community.

If this article is to be published I would hope that an honest appraisal is written into the text explaining the hydrodynamic analysis is not accurate due to the coarse scale of the model. On that basis I am unclear what value there is in undertaking the economic impacts assessment?

In summary, I do not believe the authors have addressed my initial concerns. Those elements they have addressed are flawed and in some cases inaccurate.

I maintain my initial recommendation - reject.

Reviewer #3 (Remarks to the Author):

Comments on Manuscript NCOMMS-15-22640B

The Global Flood Protection Savings Provided by Coral Reefs

The revised version of the manuscript is improved compared to the original submission, and I commend the authors for this. However, I still have a few comments about the structure of the paper and its contents.

Comments on Introduction

First, I still find the introduction unconvincing and scattered. The first paragraph in particular is not convincing. For example, the authors say that "Flooding impacts will worsen given population growth and changes in climate" (Lines 31-32). Is that the case in all locations? Aren't many places in the world (e.g. Netherlands, U.S. East Coast) preparing for this already and putting in places measures to reduce risk? Also, the authors conclude this paragraph saying that "There is an urgent need to advance risk reduction and adaptation strategies to reduce flooding impacts" (Lines 33-34). But this fact has been recognized since the 70's when the US Corps of Engineers published the first iteration of the Coastal Engineering Manual. And since then, countless publications have dealt with

this issue. So, it's unclear why the need to advance risk reduction is urgent. What is so different now from the past? And more importantly, how can this argument be used to discuss coral reefs?

I also have difficulties following the authors in pars 3-5. They seem to be presenting multiple concepts at once, and it's hard to understand the logic and order of importance of the different concepts presented. First, they say that reef cover and structure is declining (3rd par.). This leads to a claim about potential loss of benefit. But the 4th par. starts with a different claim: there's a need to quantify the benefits of reefs. This is slightly confusing, because these points are not clearly linked in the text. But then, in the same paragraph, the authors move on to another point: there's a need to do a global valuation. This last point is interesting, but it's not justified by the previous paragraph or sentences. And, it's really not clear why there's a need to do a global analysis.

But my main concern is that these 3 points do not build on each other. These are different concepts: 1- Reef structure declining, leading to a loss of services; 2- Need to value ecosystem services; 3- Need to do global analysis (even if I'm not sure why). Following these two paragraphs, the 5th par. brings an entirely new concept, which is that there's a need for better economic valuation to help better manage risk and ecosystems. Lost are the previous points, especially the one about reef decline. But more confusing is that in the same 5th paragraph, the authors say that the valuation method exists.

So, at the end of the introduction, I'm left confused because I wonder why we need another analysis, and a global one. What is so new about the current approach, if methods already exist, and what is the value added by the current analysis?

Other comments:

2nd sentence (L29-30): Simplify and clarify "discounted in development choices".

Line 39: Refs. 18 and 19 do not mention coral reefs. Why do the authors mention them?

Line 42: are storms really destructive and negative stressors to reefs? My understanding was that they can help corals reproduce because they fragment existing colonies and help them spread (see Highsmith (1982) "Reproduction by Fragmentation in Corals"). Also, if reefs protect against storms, as claimed in the ms, how can storms also be negative to their survival?

Comments on Results section

The authors are trying to make two points: 1- Reefs protect differently in different places, and 2- Reefs protect because of rugosity and height. However, I sometimes fail to understand the strength and uniqueness of the points that the authors are trying to make.

Regarding point #1, I strongly suggest that the authors improve the description of the results at a global level. In particular, I have difficulties reconciling results on Line 109 with Lines 103-105: how

does finding that reefs provide the most relative benefits to Caribbean and South Pacific regions contrasts with the result that reefs avert the most damage in Indonesia, [...] and Mexico (Line 103-105). I'm afraid that the distinction between different metrics is too subtle to really understand after a few reads. I think that's because the different points on lines 105-107; 109-111 and 113-117 all start by stating that reefs provide "flood protection benefits", but use different units. So it's hard to understand how "flood protection benefits" are measured, and it's hard to understand the point the authors are trying to make.

Regarding point #2, made in Line 105-107: Isn't this finding intuitive and obvious, given the extensive literature showing that reef rugosity and depth matters, and that reef protect against storms where there are storms in the first place? What is new about this result?

Other comments:

Line 91: Authors can improve the strength of their argument by saying more clearly that keeping corals and reefs yields to smaller losses than if all lost. The current structure of the sentence downplays the importance of reefs.

Line 96: Maybe say "At a national scale" instead of nationally?

Line 97: It is difficult understand the distinction between "for some countries" versus "for many countries". Maybe give number or percentage?

Line 98: Maybe say "for more than 200,000 people", instead of "by more ..."?

Lines 98-99: What is the approximate avoided loss value for the other countries? And the number of people protected?

Line 103: "because of reef loss" instead of "with reef loss"?

Comments on Discussion Section

Overall, I strongly suggest that the authors re-organize this section to focus on the main points. In particular, I find the discussion on uncertainty (starts at Line 142) too long and distracting. Maybe move some of it to the Methods section? More importantly, apart from the amount of effort that went into this paper (Line 152), I find the significance of the results lacking. As mentioned before, the fact that rugosity and height are important is not new (Line 121). I suggest the authors provide references where similar results have been found (see Ferrario 2014; Sheppard 2005, ...), but also explain more why the results in the present paper are so different from existing work. I also suggest that some of the content in the paragraph starting Line 152 be moved to the Methods section.

Comment on Method Section

Wave Climate (Paragraph starting line 237): It's unclear if storms are included. The datasets provided were created at such a scale that it is doubtful that they contain any hurricane information. Are they included in the tide data, if so, please clarify.

Line 271: m is the foreshore slope, not the slope from the shore to the foreshore

Line 273-274: Stockdon et al. didn't use reef profiles in their paper. So I'm confused by the distinction between setup and swash processes in this paragraph. And it's doubtful that the setup predicted by this formula compares well in reef environment, given the work that Lowe et al. did to show the importance of reef structure on setup. I strongly suggest revisiting this paragraph.

Line 275: Do waves always break offshore of the reef? What about barrier reefs, where waves break on the reef face? How does the runup compare to, say, estimates by Gourlay (1996, Coastal Engineering)?

Reviewers' comments:

Reviewer #1 (Remarks to the Author):

Thanks to the authors for this improved version. It mostly answers my concerns and I think the paper could be published as it is.

I have a couple of small suggestions:

Page 5, line 83 “reefs provide more benefits for lower intensity frequent storms”: I would add “in relative terms” since the absolute value increases in Figure 3. Also, it may be useful to add a panel to Figure 3 with the effect in percentage point, to show the decrease mentioned in the text.

We have amended as follows (note we did not use “relative” as this term was found to be confusing in review 2). Given constraints on figure space, we have put these percent changes in text.

Line 85- “For 25-year events, reefs reduce flooding for more than 8,700 km² of land and 1.7 million people, and provide \$36 billion in avoided damages to built capital (Figure 3, See SI Figure S2). For 100-year events, the topmost 1m of reefs provide flood reduction benefits that result in \$130 billion in avoided damages (Figure 3). Without reefs, damages would increase by 90% for 100-year events and a 141% for 25-year events.”

At the end of the introduction, the authors may want to add a couple of sentence recognizing that a global assessment cannot get to the level of detail that a local analysis can afford. This may help answer the concerns of the second referee.

We have added some discussion as follows.

Line 155- “Our coastal flooding analyses have several significant, combined improvements over other recent global flooding analyses^{2,5,6} including the downscaling to a 90m resolution; consideration of hydraulic connectivity in the flooding of land; the use of 30 years of wave, surge, tide and sea level data; reconstruction of the flooding height time series and associated flood return periods⁴⁹; inclusion of nearshore bathymetry and reef profiles; and the use of country-specific adjustments to allocate GDP per person. Major remaining constraints for global coastal flooding models include the consideration of flooding as a one-dimensional process and the difficulty in representing flooding well in smaller islands.

The application of a one-dimensional model neglects some of the hydrodynamics that occur on natural reefs, such as longshore flow and lagoon circulation. However, this 1-D approach is common in reef studies, either with the same wave action balance equation use here or in more complex numerical hydrodynamic studies^{15,41,45,50-53}. Flood models based on the wave action balance equation are widely employed for coastal modeling⁵⁴. The consideration of non-linear effects is only possible using phase resolving models (e.g., XBeach) at local scales (e.g., bays)^{12,15,55,56}. This modeling approach is not feasible at the global scale because of computational capacity and the lack of high resolution bathymetric data and especially if risk is to be evaluated in terms of annual expected damages. We have shown that the wave propagation approach in our global reef flooding model performs very well when considered against the results of one of these phase resolving models (see SI and Figure S7). Further the changes in flooding in our global model are consistent with changes observed in a site-validated, XBeach model that also considers flooding with changes in reef friction and sea level¹⁵.”

As a side note, I've been in the authors' shoes a few times, with reviewers doing local and high-resolution analysis expressing concerns about a global analysis that by construction cannot get to the same level of details. While I am not a specialist of the issues at stake in the comments of the second reviewer, I would like to support the authors. While local high-resolution studies are critical, they do not replace global assessments based on simpler methodologies. We do need a hierarchy of studies, from global to local scales, and from simple to complicated models.

While the global studies will have to make simplifications (for reasons related to data availability to computation requirements), they can add a lot of value in terms of research, and in practical terms they are very valuable to identify hot spots and quantify the issues. And research is iterative: the authors or other scholars (maybe the second reviewer?) may be able to start from the proposed study and improve it by adding more mechanisms and better data.

While reviewers cannot ask global studies to be as detailed as local one, they must ask authors to be very clear on the assumptions and simplifications. On this, I feel that the authors are doing a great job, and their study is very transparent.

I think the results of the sensitivity analysis should be better communicated. In particular, I could not find the impact of various assumptions on the economic assessments. It would bring a lot of confidence in the results if the authors could show that different assumptions do not completely change their results. I'm not sure how the changes in run-up would translate into changes in economic losses due to coastal floods.

We have provided extensive sensitivity analyses and have added new validation of the flooding models. We mainly report the sensitivity analysis results in terms of % change in flood height or run-up because these are the key outputs from the models. The other results for example for people and built capital are generated from GIS overlays of the flooded area.

And it would be useful to include other uncertainties – for instance the assumption that 50% of the value of a building is lost when flooded by a 1-meter flood is probably acceptable for many buildings in developing countries, it may not be the case for modern buildings (especially because many buildings in places that are regularly flooded have stilts or do not fully use the ground floor to minimize losses). The value of 50% may be even more overestimated since the “flood” is here defined by the wave height – not as the level of permanent flooding. So the authors' assumptions is different from most flood vulnerability curves, which use the permanent water level as an input (not the wave height). I would suggest to the authors to stress test their results to this assumption.

We already test sensitivity to this damage coefficient as follows.

(line 370) *“Assessing Damages and Estimating Annual Benefits. We followed existing approaches for assessing the damages to built capital as a function of the flooding level⁶. We calculated the percentage of built capital that has been damaged (D) for a given flooding level h and a certain coefficient k that must be calibrated as $D(h) = h/(h + k)$. This curve indicates that as flooding level increases, the percent of damages to built capital also increases. While there is debate about the right k to choose, we have followed others in using $k = 0.5^6$, which means that the built capital flooded at 1m of depth loses 50% of its value.”*

(Line 407) *“We also did sensitivity analyses on the damage function model with other parametrizations of k (k = 0.2, 0.3, 0.4). Lowering k lowers the total value of the built capital damaged, but has little effect on the relative effectiveness (% difference) of reefs for risk reduction. Lower k values slightly*

increase the relative effectiveness of reefs making our use of $k = 0.5$ the most conservative for comparisons.”

Reviewer #2 (Remarks to the Author):

Thank you for forwarding the revised manuscript for my reconsideration.

To answer your question directly I do not believe the amendments have adequately responded to my concerns in the revision. I highlight several specific elements in this regard.

1. Loss of reef elevation. The authors state that globally there is widespread consensus that reefs are degrading. It is true that the weight of published material does suggest that the proportion of living coral is decreasing. The authors further assert that coral degradation and structural reef loss are one in the same. This is not the case. The authors have increased the number of references to support this statement. However, they do not all measure reef loss. The Alvarez article certainly does. However, the Perry and Morgan article simply infers this might be an outcome but present no quantitative data to support this. Only include articles that actually measure structural loss. Many articles suggest this might be an outcome but fail to demonstrate this.

*We have provided the references as previously requested by the reviewers. The evidence is quite clear; reefs are dying globally and where measured the die-offs are shown to affect the height and complexity of reefs; this should not be that surprising given these die offs particularly affect shallow corals and often the large branching and boulder corals. We believe that Perry and Morgan is a valid reference as indicated by this excerpt from their abstract (we highlighted key text in **bold**).*

*“Resultant coral bleaching caused an average 75% reduction in coral cover (present mean 6.2%). Most critically we report major declines in shallow fore-reef carbonate budgets, these shifting from strongly net positive (mean 5.92 G, where G = kg CaCO₃ m⁻² yr⁻¹) to strongly net negative (mean -2.96 G). These changes have driven major reductions in reef growth potential, which have declined from an average 4.2 to -0.4 mm yr⁻¹. **Thus these shallow fore-reef habitats are now in a phase of net erosion.** Based on past bleaching recovery trajectories, and predicted increases in bleaching frequency, we predict a prolonged period of suppressed budget and reef growth states. This will limit reef capacity to track IPCC projections of sea-level rise, **thus limiting the natural breakwater capacity of these reefs and threatening reef island stability.** (Perry and Morgan 2017).*

2. I raised a specific concern related to the nature of wave breaking with reefs and the fact that across much of the Indo-Pacific the zones of critical wave breaking is depauperate of coral (comment 1c). The authors respond with:

“There is clear evidence that storms globally damage and remove massive corals including in the Indo-Pacific, which are cited in the paper). This indicates that reefs and reef height are regularly being impacted in areas critical for wave breaking and flood reduction (because wave energy is removing large corals).”

This response represents a complete lack in understanding of the nuances between individual coral growth and reef platform development and change, which is at the heart of the scenarios developed. Simply because a storm may pluck corals from the forereef does not equate to structural loss of reef surfaces. Indeed in Cyclone Bebe, extreme waves removed more than 1 million cubic metres of coral and rubble from the forereef of Funafuti atoll. This material was spread across the reef platform surface. The net effect was to increase the reef surface by >0.5 m due to the deposition of rubble as a blanket across the reef surface.

We have addressed these concerns thoroughly. We have further clarified the ‘with’ and ‘without reefs’ scenarios and indicated that (i) these scenarios are needed for a valuation, (ii) a 1m loss is conservative relative to many other examples of services assessments which assume entire loss of habitats; (iii) identified that this scenario is not meant to be a prediction of site-specific reef trajectories, and (iv) provided a clear sensitivity analysis of how changes in bathymetry can affect the results.

And we have included additional key citations on coral reef condition and reef height globally to show that there is significant loss of corals and reef substrates across reef habitats and regions.

We do not dispute that it is possible at some specific sites that reef height can increase from rubble deposition, earthquakes and even coral growth. But the preponderance of the evidence is clear from every regional or global study; reefs are dying and being eroded.

3. The authors claim the model is not of the ‘bathtub’ variety. However, I can find no element in the approach that allows for the landform feedbacks in the system that would potentially offset future flooding.

No global model of flooding, including those published recently in the *Nature* family, include landform feedbacks; it is not possible to do so globally. We do however make a major advance over most prior global flooding by advancing past bathtub models to one that uses hydraulic connectivity to flood land.

Line 352 – *“We extended the flood heights inland by ensuring hydraulic connectivity between points at a 90m resolution; a significant advance over more common bathtub approaches in earlier global flooding models. From the flooding levels and flooding extent, we calculated the total area of land affected and damages at each study unit. Flooding maps were also intersected with population data⁶⁷ after resampling from the original 1 km resolution to the 90 m of the digital elevation model.”*

4. Hydrodynamic modelling. The authors appear to agree with my concerns related to hydrodynamic modelling. They state that the necessary level of analysis to fully understand wave reef interactions "...is only possible at local scales (e.g. bays) using phase resolving models that require high resolution bathymetry and large computational efforts."

The argument being put forward seems to rest on the global scale of the model, which the authors acknowledge is at too coarse a scale to represent wave-reef-coast interactions very well. Furthermore, the authors appear to argue that this model approach is an improvement on other approaches.

I disagree that the paper should be evaluated relative to other large scale flooding models, as asserted by the authors. The substance of the article will have significant international traction. The reef community deserves outcomes that can be relied upon at the reef scale. The admission that the model cannot yield this - would seem to be a fatal flaw.

I fully understand the attractiveness of attempting to value what coral reefs can afford, from multiple angles. However, I am unconvinced that a better, but wrong answer (which the authors acknowledge, due to the computational complexities), is something that should be released to the broader science community.

If this article is to be published I would hope that an honest appraisal is written into the text explaining the hydrodynamic analysis is not accurate due to the coarse scale of the model. On that basis I am

unclear what value there is in undertaking the economic impacts assessment?

We appreciate that the reviewer now notes “The substance of the article will have significant international traction.” It has been our experience that many scientists and leaders are keenly interested in these results, because they have never seen anything like them before at the scale and rigor that we provide.

We now show more clearly that the models that we use at the global scale follow the same or similar approaches that are commonly used for site-specific reef studies – i.e., at the “reef scale”. Further we provide a direct comparison between our model and a more complex, site-specific model of the type called for in review 2. We show that our model performs well and that the results are consistent with other published papers that also rely on similar cross-shore profile approaches (i.e., 1-D) to study the hydrodynamics of coral reefs at sites.

We address these concerns as follows;

Line 155 – “Our coastal flooding analyses have several significant, combined improvements over other recent global flooding analyses^{2,5,6} including the downscaling to a 90m resolution; consideration of hydraulic connectivity in the flooding of land; the use of 30 years of wave, surge, tide and sea level data; reconstruction of the flooding height time series and associated flood return periods⁴⁹; inclusion of nearshore bathymetry and reef profiles; and the use of country-specific adjustments to allocate GDP per person. Major remaining constraints for global coastal flooding models include the consideration of flooding as a one-dimensional process and the difficulty in representing flooding well in smaller islands.

The application of a one-dimensional model neglects some of the hydrodynamics that occur on natural reefs, such as longshore flow and lagoon circulation. However, this 1-D approach is common in reef studies, either with the same wave action balance equation use here or in more complex numerical hydrodynamic studies^{15,41,45,50-53}. Flood models based on the wave action balance equation are widely employed for coastal modeling⁵⁴. The consideration of non-linear effects is only possible using phase resolving models (e.g., XBeach) at local scales (e.g., bays)^{12,15,55,56}. This modeling approach is not feasible at the global scale because of computational capacity and the lack of high resolution bathymetric data and especially if risk is to be evaluated in terms of annual expected damages. We have shown that the wave propagation approach in our global reef flooding model performs very well when considered against the results of one of these phase resolving models (see SI and Figure S7). The changes in flooding in our global model also are consistent with changes observed in a site-validated, XBeach model that also considers flooding with changes in reef friction and sea level¹⁵.”

Supplementary Information

*Line 308 – “**Validation of the wave and flooding model.** Our physical modeling approach relies on linear wave theory to calculate the effect of reefs on waves and wave setup. We use a modification of an empirical formula to estimate the swash component of run-up (see above) that was originally calculated for beach environments. Models based on the same equation (e.g., SWAN) have been used to study the effect of reefs on flooding at regional scales^{28,65,66}. More complex but still 1D models have been used for the study of the hydrodynamics of coral reefs in more detail at specific sites and they also use transect-based approaches⁴².*

We compare and validate our modeling approach against a more complex model for reef environments, XBeach^{43,67} in a theoretical fringing reef profile. The comparison was performed in a frictionless setting to focus on comparisons of wave breaking and run-up. XBeach was originally derived from sandy beaches and has

hitherto been successfully applied to predict erosion and overwash under hurricane forcing⁶⁸ and more recently, successfully applied in reef environments^{42,43}.

We considered three different reef depths or sea levels (0, 1.5 and 3m) and three different offshore significant wave heights (1, 3 and 6m) and estimated flood height at the shore with each model. The comparison shows that the flood height values of our approach are comparable to those from a higher resolution model (Figure S7, $r^2=0.978$) given the simple geometries that are characteristic of the global bathymetric data.“

Supplementary Figure S7: Comparison of flood height at the shore predicted by XBeach and the global model in this paper. We examined results for a fringing reef profile under 3 different reef depths (0, -1.5m, - 3m) represented by the different colors and three different significant wave heights ($H_s= 1, 3$ and $6m$).

In summary, I do not believe the authors have addressed my initial concerns. Those elements they have addressed are flawed and in some cases inaccurate.

I maintain my initial recommendation - reject.

Reviewer #3 (Remarks to the Author):

Comments on Manuscript NCOMMS-15-22640B
The Global Flood Protection Savings Provided by Coral Reefs

The revised version of the manuscript is improved compared to the original submission, and I commend the authors for this. However, I still have a few comments about the structure of the paper and its contents.

Comments on Introduction

First, I still find the introduction unconvincing and scattered. The first paragraph in particular is not convincing. For example, the authors say that “Flooding impacts will worsen given population growth

and changes in climate” (Lines 31-32). Is that the case in all locations? Aren’t many places in the world (e.g. Netherlands, U.S. East Coast) preparing for this already and putting in places measures to reduce risk? Also, the authors conclude this paragraph saying that “There is an urgent need to advance risk reduction and adaptation strategies to reduce flooding impacts” (Lines 33-34). But this fact has been recognized since the 70’s when the US Corps of Engineers published the first iteration of the Coastal Engineering Manual. And since then, countless publications have dealt with this issue. So, it’s unclear why the need to advance risk reduction is urgent. What is so different now from the past? And more importantly, how can this argument be used to discuss coral reefs?

The citations provided in this paragraph (including Nature Climate Change papers) provide very clear evidence for the statements we make; flood risks are rising and must be addressed. We agree that the problem of flooding is not new but there are widespread concerns globally about the problem (they are regularly discussed in popular media).

I also have difficulties following the authors in pars 3-5. They seem to be presenting multiple concepts at once, and it’s hard to understand the logic and order of importance of the different concepts presented. First, they say that reef cover and structure is declining (3rd par.). This leads to a claim about potential loss of benefit. But the 4th par. starts with a different claim: there’s a need to quantify the benefits of reefs. This is slightly confusing, because these points are not clearly linked in the text. But then, in the same paragraph, the authors move on to another point: there’s a need to do a global valuation. This last point is interesting, but it’s not justified by the previous paragraph or sentences. And, it’s really not clear why there’s a need to do a global analysis.

But my main concern is that these 3 points do not build on each other. These are different concepts: 1- Reef structure declining, leading to a loss of services; 2- Need to value ecosystem services; 3- Need to do global analysis (even if I’m not sure why). Following these two paragraphs, the 5th par. brings an entirely new concept, which is that there’s a need for better economic valuation to help better manage risk and ecosystems. Lost are the previous points, especially the one about reef decline. But more confusing is that in the same 5th paragraph, the authors say that the valuation method exists. So, at the end of the introduction, I’m left confused because I wonder why we need another analysis, and a global one. What is so new about the current approach, if methods already exist, and what is the value added by the current analysis?

We have made some revisions to the introduction though we also believe that these points represent editorial differences. The topic sentences for every paragraph in the introduction are below; we believe that these present a clear and coherent argument and that the paragraphs back up these points.

1. *“The economic impacts of coastal flooding are substantial^{1,2}.”*

2. *“Coral reefs serve as natural, low-crested, submerged breakwaters, which provide flood reduction benefits through wave breaking and wave energy attenuation.”*

3. *“Reefs have experienced significant losses globally in living corals and reef structures from coastal development; sand and coral mining; overfishing and destructive (e.g., dynamite) fishing; storms; and climate-related bleaching events^{8,20-23}.”*

4. *“Although reefs and other coastal habitats can provide flood protection benefits, they are rarely accounted for directly in coastal management, because these services are not quantified in terms familiar to decision-makers, such as (loss of) annual expected benefits¹².”*

5. *“Better valuations of the protection services from coastal habitats could inform decisions to meet multiple objectives in risk reduction and environmental management³⁵⁻³⁸.”*

6. *“Natural flood protection benefits are amenable to spatially-explicit quantification, because of the broader work on assessments of flood risks and artificial coastal defenses.”*

7. *“Using process-based flooding models, we estimate the annual expected benefit of coral reefs for protecting people and property globally.”*

Other comments:

2nd sentence (L29-30): Simplify and clarify “discounted in development choices”.

Instead of *“these risks are being excessively discounted in development choices”* we now more simply say *“these risks are often discounted in development choices”*. We believe it was the term “excessively”, which was most confusing.

Line 39: Refs. 18 and 19 do not mention coral reefs. Why do the authors mention them?

We include these references because the sentence points to “reefs and other coastal habitats”

“The flood reduction benefits of coral reefs and other coastal habitats are predicted to be high and even cost effective in comparison to traditional approaches^{13,17-19}.”

Ref 18 only covers wetlands, but ref 19 does cover coral reefs.

Line 42: are storms really destructive and negative stressors to reefs? My understanding was that they can help corals reproduce because they fragment existing colonies and help them spread (see Highsmith (1982) “Reproduction by Fragmentation in Corals”). Also, if reefs protect against storms, as claimed in the ms, how can storms also be negative to their survival?

It is appropriate to include storms as a factor that impact reefs and we think it would be an oversight not to. We state *“Reefs have experienced significant losses globally in living corals and reef structures from coastal development; sand and coral mining; overfishing and destructive (e.g., dynamite) fishing; storms; and climate-related bleaching events^{8,20-23}.”*

References such as De’ath et al (ref 22, PNAS ‘12) clearly show that it is critical to consider impacts from storms (excerpt from their abstract below- bold text ours)–

*“Based on the world’s most extensive time series data on reef condition (2,258 surveys of 214 reefs over 1985–2012), we show a major decline in coral cover from 28.0% to 13.8%(0.53%y⁻¹), a loss of 50.7% of initial coral cover. **Tropical cyclones, coral predation by crown-of-thorns starfish (COTS), and coral bleaching accounted for 48%, 42%, and 10% of the respective estimated losses**, amounting to 3.38% y⁻¹ mortality rate.”*

Comments on Results section

The authors are trying to make two points: 1- Reefs protect differently in different places, and 2- Reefs protect because of rugosity and height. However, I sometimes fail to understand the strength and uniqueness of the points that the authors are trying to make.

We focus on the most rigorous sub-national, national and global values of **the annual expected social and economic benefits of reefs**; these results are unique and there are not any other papers close to ours in the rigor and geographic scale of the estimation of a marine ecosystem service.

Regarding point #1, I strongly suggest that the authors improve the description of the results at a global level. In particular, I have difficulties reconciling results on Line 109 with Lines 103-105: how does finding that reefs provide the most relative benefits to Caribbean and South Pacific regions contrasts with the result that reefs avert the most damage in Indonesia, [...] and Mexico (Line 103-105). I'm afraid that the distinction between different metrics is too subtle to really understand after a few reads. I think that's because the different points on lines 105-107; 109-111 and 113-117 all start by stating that reefs provide "flood protection benefits", but use different units. So it's hard to understand how "flood protection benefits" are measured, and it's hard to understand the point the authors are trying to make.

We have clarified further that at a national level you can consider impacts in

1. Absolute terms – e.g., total dollar value of national avoided losses from reefs.
2. Relative terms – e.g., total dollar value of national avoided losses from reefs per GDP (i.e., relative to the size of the national economy).

We have clarified as follows.

Line 103 "The national benefits of reefs for flood protection can be considered not just in total built capital and people protected but also relative to the size of the national economy (Table 1) and population (Tables S1, S2). These results highlight the importance of reefs to many smaller island nations in the Caribbean and the South Pacific, which receive significant benefits relative to their Gross Domestic Product (GDP) (Table 1). The flood protection benefits of coral reefs are particularly critical in the Philippines, Malaysia, Cuba and the Dominican Republic (Table 1). In these countries, reefs are important for averting damages both to built capital overall (total dollar value of national avoided losses) and relative to the size of their economies (i.e., total dollar value of national avoided losses/GDP)."

We do not believe that this is a subtle difference and it fundamentally underlies for example why nations globally care about small, developing nations with regards to flooding and climate change. For example, the impacts of flooding on the Maldives are frequently discussed in popular media such as the NY Times (e.g., March 30, 2017) not because of the absolute number of people impacted (very small in total numbers) but because a huge amount of the country will be affected in relative terms (potentially 100% of the national population).

Regarding point #2, made in Line 105-107: Isn't this finding intuitive and obvious, given the extensive literature showing that reef rugosity and depth matters, and that reef protect against storms where there are storms in the first place? What is new about this result?

"Overall the hotspot areas where reefs provide the most flood protection benefits are in storm belts with extensive, shallow and rugose coral reefs; land at low elevation; and assets concentrated on the coast (Figure 5)."

We agree that this point is clear and obvious. We are stating the main factors that underlie our findings; it is a relevant statement for the results. Further we do not think this point is actually made very often because there are very few analyses that quantitatively consider these factors together- and none anywhere close to the scale we consider.

Other comments:

Line 91: Authors can improve the strength of their argument by saying more clearly that keeping corals and reefs yields to smaller losses than if all lost. The current structure of the sentence downplays the importance of reefs.

We have added to the point as suggested.

“Future sea level rise will increase risks, and these risk will be even greater if reefs are lost too (Figure 4).”

Line 96: Maybe say “At a national scale” instead of nationally?

We made this change.

Line 97: It is difficult understand the distinction between “for some countries” versus “for many countries”. Maybe give number or percentage?

We have clarified as follows...

Line 96- “At a national scale, reefs provide annual expected benefits of hundreds of millions of dollars in avoided flood damages for five countries and millions of dollars in annual benefits for more than 20 additional countries (Table 1).”

Line 98: Maybe say “for more than 200,000 people”, instead of “by more ...”?

In the previous reviews, reviewer 1 explicitly asked that we do the opposite (say “by more” instead of “for more”). We feel that both actually mean the same thing.

Lines 98-99: What is the approximate avoided loss value for the other countries? And the number of people protected?

All of these values are expressed in the tables cited in this paragraph (i.e., Tables 1 & S1).

Line 103: “because of reef loss” instead of “with reef loss”?

We have revised this sentence to focus on current benefits.

Line 113 “The places where reefs avert flood damages to people are more widespread geographically (See SI, Figure S1),”

Comments on Discussion Section

Overall, I strongly suggest that the authors re-organize this section to focus on the main points. In particular, I find the discussion on uncertainty (starts at Line 142) too long and distracting. Maybe move some of it to the Methods section? More importantly, apart from the amount of effort that went into this paper (Line 152), I find the significance of the results lacking. As mentioned before, the fact that rugosity and height are important is not new (Line 121). I suggest the authors provide references where similar results have been found (see Ferrario 2014; Sheppard 2005, ...), but also explain more why the results in the present paper are so different from existing work. I also suggest that some of the content in the paragraph starting Line 152 be moved to the Methods section.

We appreciate these comments and we have done a better job of highlighting the key results up front. We agree with the review that some of these core points got buried behind the many other points we were asked in prior reviews to address in the discussion.

We agree that the discussion on uncertainty is still longer than that required for other global flooding analyses, but we have been asked to provide this level of detail in response to the other reviews.

We are much clearer on the novel findings and the rigorous and direct valuation of social and economic benefits. We note that reef height and rugosity are important factors in estimating this benefit, but they are only one part of the extensive data gathering and modeling required to achieve the key results in this paper.

Comment on Method Section

Wave Climate (Paragraph starting line 237): It's unclear if storms are included. The datasets provided were created at such a scale that it is doubtful that they contain any hurricane information. Are they included in the tide data, if so, please clarify.

Storms are included in the wave climate data, but in some areas the effects of hurricanes are underestimated. This has been extensively analyzed by some of our co-authors (see Figure 5 in Reguero et al. 2015). In text we note that

Line 245 "Storms are generally captured well in these wave data though the wave heights in some of the hurricane events can be underrepresented^{61,72}."

Line 271: m is the foreshore slope, not the slope from the shore to the foreshore

We have extensively revised these sections to clarify these models and the parameter estimation as indicated fully in our response in the next comment.

Line 273-274: Stockdon et al. didn't use reef profiles in their paper. So I'm confused by the distinction between setup and swash processes in this paragraph. And it's doubtful that the setup predicted by this formula compares well in reef environment, given the work that Lowe et al. did to show the importance of reef structure on setup. I strongly suggest revisiting this paragraph.

We agree that Stockdon does not use reef profiles. We identified an approximation for assessing run-up in reef environments using Stockdon (starts line 266). This approach was the best given the existing bathymetric data. We have now added a validation of this approach against a more complex non-linear wave model and showed that this approximation is reasonable (and added to methods and Supplementary Information- also noted above). We also make very clear in the Discussion that approaches like that proposed by Lowe are not feasible beyond a few locations because of the lack of high resolution bathymetric data (relevant for all flooding models) and computational capacity.

Below we have provided the revised sections from the Methods in the mss. The methods in the Supplementary Information go in to even further detail.

Line 258- **“Nearshore hydrodynamics: (b) reef wave model.** Wave propagation over the reef is calculated from linear wave theory. Wave propagation is modeled at shore-perpendicular one-dimensional transects therefore processes such as longshore currents, are neglected. The evolution of a wavefield of root-mean square (rms) wave height H with weak mean currents is computed by solving the wave energy balance equation:

$$\frac{\partial E_w C_g}{\partial x} = -(D_b + D_f + D_v) \quad (1)$$

where E_w is the wave energy density and C_g the group velocity. The dissipation of wave energy flux is caused by wave breaking (D_b), bottom friction (D_f), and the presence of vegetation in the water column (D_v), which is not considered in this study. Equation (1) is widely applied in coastal studies to assess wave propagation (e.g. SWAN)⁷⁷ and previously applied to reef environments⁵⁰. D_b and D_f are expressed following Thornton and Guza⁷⁸:

$$D_b = \frac{3\sqrt{\pi}}{16} \cdot \rho \cdot g \cdot \frac{B^3 \cdot f_p}{\gamma^4 \cdot h^5} \cdot H^7 \quad (2)$$

$$D_f = \frac{f_w}{16\sqrt{\pi}} \left(\frac{\sigma}{\sinh(kh)} \right)^3 \cdot H^3 \quad (3)$$

where ρ is water density, g the constant of gravity, k the wave number, σ the wave frequency and f_p the peak frequency. The breaking coefficient B and breaker index γ have the default values of 1.0 and 0.78 and the bottom friction coefficient f_w is taken as 0.01 for sand beds^{64,78}.

In our model, we implement recent studies on wave transformation by coral reefs^{16,79} and replace the breaker index (B) by an expression where h/H provides the relationship between water depth and wave height at breaking conditions:

$$\gamma_{coral} = 0.23 \cdot \tanh \left[2.3143 \cdot \left(1.4 - \frac{h}{H} \right) + 3.6522 \right] \quad 0 < \frac{h}{H} < 2.8 \quad (4)$$

Nearshore hydrodynamics: (c) total water level model. The total water level (i.e., flood height) along shorelines is a function of mean sea level, astronomical tide, storm surge, and the run-up of waves⁸⁰. The run-up represents the wave-induced motion of the water's edge across the shoreline and is built of two contributions, namely the wave setup at the shoreline and the swash representing oscillations about the setup. The run-up calculation requires obtaining the local wave conditions at the shoreline using the reef wave model above.

Nearshore hydrodynamics: (d) computation of wave setup. The wave-setup can be obtained from the conservation of mass and the momentum equations⁸¹. In our one-dimensional setting, the computation of the wave-induced setup is based on the vertically integrated momentum balance equation⁸². Similar implementations have been used in previous work to evaluate the effect of vegetation on wave-induced setup⁴⁵ and in coral reef environments⁸³.

Nearshore hydrodynamics: (e) computation of wave runup. The 2% exceedance level of wave runup maxima generated by random wave fields on open coast sandy beaches was estimated in Stockdon et al⁸⁴ as:

$$R_{u,Stockdon} = 1.1 \cdot \left(0.35m\sqrt{H_0 L_0} + \frac{\sqrt{0.004H_0 L_0 + 0.563H_0 L_0 m^2}}{2} \right) = 1.1 \cdot \left(\bar{\eta} + \frac{\sqrt{S_{inc}^2 + S_{if}^2}}{2} \right) \quad (5)$$

where H_0 is the offshore significant wave height, L_0 represents the deep-water wave length, T_p the peak period, and m the bathymetry slope in the foreshore beach slope. This equation expresses runup as a function of empirical estimates of incident wave setup at the shoreline $\bar{\eta}$ and the swash incident and infragravity band frequency band components (S_{inc} , S_{if} , respectively).

For the swash component (i.e., the second term in the equation), there is no specific formulation for reef environments. Here, we apply a modification of this second term to find the contribution of the swash to the total water level. First we calculate the breaking point (h_b) from the reef wave model and the dissipation of waves to define the surf zone from that point to the shore; (ii) then in the 'surf zone', we calculate the equivalent slope, m^* , as if it were a beach profile; and (iii) using slope m^* , we calculate the offshore wave height, H_0^* , which will give this same wave breaking point (h_b), and we use this H_0^* in equation 5 (see SI). This modification assumes that the processes in the surf zone that control wave runup are similar in our reef profiles to a beach profile with similar surf zone slope. Modifications of the same formula have been applied previously to estimate the effect of vegetated ecosystems on runup⁴⁵.

Extreme water levels and flood height time series reconstruction

From the propagations of waves and the calculation of total water levels onshore (above), the reconstruction of the flood height time series at the most onshore points is based on multi-dimensional interpolation techniques⁶⁶. We apply a Peak Over Threshold method to select extreme flood heights and fit a General Extreme Value Distribution⁸⁵ to obtain the flood heights associated with the 10-, 25-, 50- and 100-year return periods. The methodology has been tested in case studies and validated with observations^{86,87}.”

Line 275: Do waves always break offshore of the reef? What about barrier reefs, where waves break on the reef face? How does the runup compares to, say, estimates by Gourlay (1996, Coastal Engineering)?

Waves do not always break offshore of the reef. We find the breaking point of waves from the propagation of each sea state and for each profile (see Methods in prior comment). The breaking points depend on wave height, periods, surge, mean sea level and geometry of the profile. Thus some breaking point occur offshore of the reef, others inshore, as determined by the site-specific geometry and sea states.

From the wave propagation, we calculate the setup, consistent with Gourlay 1996a,b from the propagation of each sea state, and then apply the runup equation from each breaking point shorewards (after the breaking). The application of the runup formula, originally parameterized for dissipative beaches, is only applied in the surf zone (after the breaking point), assuming the reef runup zone resembles the runup in an equivalent beach with similar slope (see above). We have validated this approach with simulations in XBeach as noted in previous responses.

Reviewers' comments:

Reviewer #1 (Remarks to the Author):

Thanks for the revised version of the paper.

I like the paper, which is clear and well-written, and very open on the limits of the analysis.

I find the criticisms from reviewer #2 misguided. One cannot apply the same evaluation criteria to a global and a local study. And it is well documented how global and local studies build on each other and contribute to better knowledge. All global studies can be accused of not considering enough the local details (the same can be said of all local studies). But all local studies can be said to lack broader significance beyond the considered system. Only a combination of the two approaches can provide us with the information we need to better understand the issues at stake. So I would favor publication of the paper.

Maybe an addition that could satisfy the criticism of reviewer #2 is a more systematic comparison of the approach used at the global scale and the more sophisticated analyses possibly at the local level. One could imagine a table with a row per "module" of the model (the subsections of the Method section), and one column for the global model used here, and one column for the "best practice" in local analysis. (And possibly a column what what's not included or represented in local studies.) Such a table would clearly show the limitations of the paper, but would also be very useful for the community, flagging where progress is still needed.

Reviewer #2 (Remarks to the Author):

This is the third time I have reviewed this manuscript. I have raised a number of concerns with this paper in past responses. Some of these concerns have been addressed by the authors.

In response to the above questions I am not convinced it will influence thinking in the field. The global community is convinced of the value of reefs as coastal protection structures that mitigate impacts on coastal communities. This paper reinforces this widely held belief.

More specifically, I am pleased to see the manuscript contains a more nuanced discussion of the strengths and weaknesses of the modelling approach. This provides clarity for the reader in how realistic the results are. I accept they have now highlighted the weaknesses in the manuscript. It would be nice to see further discussion on what these weaknesses might imply for site specific analyses.

I believe there are two broader issues that need to be refined.

1. It is still asserted that there is a global reef flattening (line and repeated in line 181). This is simply not the case. There are a few studies that report this in isolated places. There are still many more locations where reef flattening has not been observed and indeed reefs are still accreting. The Perry paper used to support this statement does not even examine reef top accretion. It examined the health of the reef in isolated locations on the forereef slope. The ecological conditions of this zone are distinctly different to the coral-algal reef rim...which is still healthy and accreting. A close reading of the Sheppard et al paper also shows they did not measure structural loss, but rather inferred such changes.

I think the solution here is to note that reef response will be spatially variable. In some places there is, and may be, structural loss.....in others there is not. Consequently, the flood risk problem will scale dependent on the likely reef trajectory. I think this provides a more interesting discussion element than a blanket rule about future responses. It is grossly misleading to indicate reef flattening is a global phenomenon.

Line 179. A 1m change in reef profile is not modest.

2. Management implications

I understand the rationale of the manuscript and consequently the call for improved management of reefs to offset damages to communities. However, surely the manuscript also needs to acknowledge that an equally sensible strategy to avoid flood impacts is to reconsider how communities have developed and settled the low-lying coastal fringe? How should they adapt??

Reviewer #4 (Remarks to the Author):

I have carefully read the revised paper, the reviewers' comments and the authors' responses to the comments. I am finding myself agreeing with Reviewer 2 and see limited use of this global approach to quantifying the value of coral reef systems for coastal protection. I agree that the approach is too crude to deliver meaningful results and I am also somewhat puzzled by the methodology. For example, wave transformation is carried out, but Stockdon equation uses deep water wave conditions, and was developed for 'planar' beach profiles. Additionally, I do not see proper consideration of the water levels and how, say, the 1:100 year wave condition is associated with the water level. Wouldn't this warrant some sort of joint probability approach? I think the problem here is that, due to this being a Nature group publication, the methodology is very much truncated, making it difficult for the reader to fully comprehend the approach followed and the assumptions underpinning this modelling exercise.

With respect to the comments of Reviewer 3, the authors appear to have responded to most comments, but, really, Reviewer 3 should be asked whether they are happy with the response. I fully agree with Reviewer 3 that the conclusion 'Overall the hotspot areas where reefs provide the most flood protection benefits are in storm belts with extensive, shallow and rugose coral reefs; land at low elevation; and assets concentrated on the coast' is profoundly unsurprising.

I see the point made by Reviewer 1 that one can always criticise the global approach as being too simplistic and that there is a place for such approaches. But, is the Nature Group the correct outlet for such publications? Based on past evidence it clear is, as there are a plethora of such global-reach papers. Generally, they are not particularly rigorous, nor very insightful in terms of providing novel insights; however, they do make for good media soundbites and that is what seems to drive these publications: citability rather than scientific rigour and/or novelty. I believe this is an editorial decision, but in my view the quality of this paper does not meet the high standards I would expect of a Nature group publication.

Reviewers' comments:

Reviewer #1 (Remarks to the Author):

Thanks for the revised version of the paper.

I like the paper, which is clear and well-written, and very open on the limits of the analysis.

I find the criticisms from reviewer #2 misguided. One cannot apply the same evaluation criteria to a global and a local study. And it is well documented how global and local studies build on each other and contribute to better knowledge. All global studies can be accused of not considering enough the local details (the same can be said of all local studies). But all local studies can be said to lack broader significance beyond the considered system. Only a combination of the two approaches can provide us with the information we need to better understand the issues at stake. So I would favor publication of the paper.

Maybe an addition that could satisfy the criticism of reviewer #2 is a more systematic comparison of the approach used at the global scale and the more sophisticated analyses possibly at the local level. One could imagine a table with a row per "module" of the model (the subsections of the Method section), and one column for the global model used here, and one column for the "best practice" in local analysis. (And possibly a column what what's not included or represented in local studies.) Such a table would clearly show the limitations of the paper, but would also be very useful for the community, flagging where progress is still needed.

We appreciate the suggestion from Review 1, and we have provided the Table S4 in the Supplementary Information (line 408) to compare these global and local approaches (this table is also provided further below). We agree that no global study would ever be published if it had to adhere to the type of analyses that could only be done at sites with high resolution data and models.

Reviewer #2 (Remarks to the Author):

This is the third time I have reviewed this manuscript. I have raised a number of concerns with this paper in past responses. Some of these concerns have been addressed by the authors.

In response to the above questions I am not convinced it will influence thinking in the field. The global community is convinced of the value of reefs as coastal protection structures that mitigate impacts on coastal communities. This paper reinforces this widely held belief.

We disagree that the global community is generally well aware of these services. We note that none of the reviews has ever pointed to any study like ours; a geographically broad set of ecological, engineering, and economic models, which provides annual expected benefits for habitats. There are only a handful of site-based studies of reef coastal protection (we cite them). None of these site-based papers provides a probabilistic assessment of economic risk. Our assessment of annual expected benefits represents a major advance at any scale.

We also wholeheartedly disagree that this work will not influence thinking in the field. We have presented these results widely to leading scientific organizations (e.g., Pew Marine Fellows), development banks (World Bank, German Development Bank), management agencies (e.g., FEMA, USACE), and insurance companies (e.g., Lloyd's, Munich re, Swiss re). In all cases, they have indicated that nothing like our study exists, and they are keen to use the results when they can be made available.

More specifically, I am pleased to see the manuscript contains a more nuanced discussion of the strengths and weaknesses of the modelling approach. This provides clarity for the reader in how realistic the results are. I accept they have now highlighted the weaknesses in the manuscript. It would be nice to see further discussion on what these weakness might imply for site specific analyses.

We appreciate the recognition that we have gone to extensive lengths to discuss the benefits and limitations of these global approaches in text and with sensitivity analyses. We have now gone further in addressing them in Table S4 and associated text. We also note that this review has highlighted a major strength in our analyses in the

probabilistic assessment of economic risk (e.g., annual expected benefits), which has not been done at any scale for reefs or mangroves.

I believe there are two broader issues that need to be refined.

1. It is still asserted that there is a global reef flattening (line and repeated in line 181). This is simply not the case. There are a few studies that report this in isolated places. There are still many more locations where reef flattening has not been observed and indeed reefs are still accreting. The Perry paper used to support this statement does not even examine reef top accretion. It examined the health of the reef in isolated locations on the forereef slope. The ecological conditions of this zone are distinctly different to the coral-algal reef rim... which is still healthy and accreting. A close reading of the Sheppard et al paper also shows they did not measure structural loss, but rather inferred such changes.

At this point, we have added significant references and defended all of our statements about reef degradation and loss. Our points are completely in line with the vast majority of the reef literature. It is true that there is variability in these patterns but the overwhelming signal is clear particularly in areas with significant human development. Review 2 is right that there could be more direct measures of reef height. But there are very many measures of degradation all pointing to a clear and widespread pattern of reef loss.

We will also reiterate that our scientific analysis does not assume either what is happening or will happen to reefs. Our analysis only provides a 'with' and 'without' reefs scenario as required by any valuation of ecosystem services and benefits.

We understand that even clear patterns do not mean loss is happening everywhere, and we have acknowledged this fact (see text in italics below). The considerations of reef loss are solely in the Introduction and Discussion; we believe that we have done our due diligence in defending these points and respectfully request that any further points must be considered as differences in opinion about the available literature.

I think the solution here is to note that reef response will be spatially variable. In some places there is, and may be, structural loss.....in others there is not. Consequently, the flood risk problem will scale dependent on the likely reef trajectory. I think this provides a more interesting discussion element than a blanket rule about future responses. It is grossly misleading to indicate reef flattening is a global phenomenon.

As requested we have provided even more indication that there is spatial variability in the reef response.

Line 174 "This flattening of coral reefs has been observed globally^{14,25,26} and can be accelerated by coral bleaching, as witnessed during the 2015 El Niño. In the long term, these effects could be coupled with flooding impacts from a 1 m or more rise in sea levels⁵ and lead to compounding effects later in the century. *However these effects are not foregone conclusions and in some areas reefs are still in good condition and even growing. The challenge will be to maintain, improve and restore healthy reefs, which will likely require more innovative effort in the areas where the protection benefits are greatest, i.e., directly adjacent to populated areas. Better decisions in coastal development and habitat restoration could reduce risks to both people and reefs.*"

Line 179. A 1m change in reef profile is not modest.

Our study is a valuation of ecosystem services. Most of these prior studies assume the entire loss of habitats (i.e., a comparison scenario with absolutely no reefs), which means that most other studies assume meters to tens of meters of reef loss. Thus a change of only 1m in a "no reef" scenario is actually quite conservative. We have clearly noted this in text.

2. Management implications

I understand the rationale of the manuscript and consequently the call for improved management of reefs to offset damages to communities. However, surely the manuscript also needs to acknowledge that an equally sensible strategy to avoid flood impacts is to reconsider how communities have developed and settled the low-lying coastal fringe? How should they adapt??

Our first paragraph of the introduction already noted the cumulative risks of coastal development, and we have sharpened this text further (Line 29). We have also made a very clear point about risks and coastal development in the discussion (Line 177). We agree wholeheartedly with this point of view that the risks from poor development choices are clear. However we do not think that in this paper on reef benefits that it would be appropriate to dwell further on the hundreds of different choices in coastal development; we believe that in the very limited discussion space that we should focus on those most directly at the interface between reefs and people.

Line 29 *“The impacts of coastal flooding are substantial and growing as the exposure of assets increases^{1,2}. Unfortunately, these risks are often discounted in development choices^{3,4}. Flooding impacts will worsen given population growth, coastal development and climate change^{2,5,6}. Coastal development also causes losses in coastal habitats, which will further heighten risks⁷⁻¹⁰. There is a pressing need to advance risk reduction and adaptation strategies to reduce flooding impacts^{4,6,11}. “*

Line 177 *“However these effects are not foregone conclusions and in some areas reefs are still in good condition and even growing. The challenge will be to maintain, improve and restore healthy reefs, which will likely require more innovative effort in the areas where the protection benefits are greatest, i.e., directly adjacent to populated areas. Better decisions in coastal development could reduce risks to both people and reefs.”*

Reviewer #4 (Remarks to the Author):

I have carefully read the revised paper, the reviewers' comments and the authors' responses to the comments. I am finding myself agreeing with Reviewer 2 and see limited use of this global approach to quantifying the value of coral reef systems for coastal protection. I agree that the approach is too crude to deliver meaningful results and I am also somewhat puzzled by the methodology. For example, wave transformation is carried out, but Stockdon equation uses deep water wave conditions, and was developed for 'planar' beach profiles. Additionally, I do not see proper consideration of the water levels and how, say, the 1:100 year wave condition is associated with the water level. Wouldn't this warrant some sort of joint probability approach?

We have thoroughly defended our approaches in text and sensitivity analyses. We remain a little surprised that our models are not considered relative to other global flooding models published recently in *Nature* journals. And that the reviews appear to widely accept the coastal protection benefits of reefs despite the fact that there are only a few site-based studies and none studies that quantify these social or economic benefits.

As explained in methods, we have modified Stockdon to factor in reef effects. We have clarified further in the methods that their model was developed in part for barred beaches, which resemble coral reef protected beaches. Line 312 – *“In our approach, we assume that Stockdon et al.⁴¹ can be applied to coral reefs as the model was developed to include barred beaches, which resemble coral reef protected beaches.”*

With regard to the consideration of water level, we work with the total water level to estimate flooding. Total Water Level (TWL) combines effects of significant wave height, storm surge, and other sea level components as detailed in the methods. We have reconstructed three decades of the TWL time series to obtain the extreme value distributions of TWL and thus flood levels (including for example the 1 in 100 year flood). We do not need to calculate the joint distribution of the components, because they are all included in the TWL statistics.

I think the problem here is that, due to this being a Nature group publication, the methodology is very much truncated, making it difficult for the reader to fully comprehend the approach followed and the assumptions underpinning this modelling exercise.

Following the recommendations in reviews 1 and 3, we clarified the models even further in text and most importantly have provided Table S4 (Supplementary information Line 408 and below), which includes a summary of the data and models that we have used and compares them to site-based models. Previously we provided a direct comparison between our model and site-based models. We show that our model performs well and that the results are consistent with other published papers that also rely on similar cross-shore profile approaches (i.e., 1-D) to study the hydrodynamics of coral reefs at sites.

Supplementary Table S4 | A comparison of key models, data, benefits and limitations between the global approaches in Beck et al. and the approaches possible in local or site-based studies.

		Global Study (~1,000km)		Local or Site-Based Study (~1km)	
		Key Models & Data	Benefits & Limitations	Key Models & Data	Benefits & Limitations
1	Offshore hydrodynamics	 • Storm Surge (Global every 2° from 1891-2010) • Waves from reanalysis (Global data 0.25° from 1979-2017) • Astronomical Tide (1°, altimetry) • Local relative sea level rise projections • Global land subsidence • Reanalysis or hindcasts do include validation and calibration with observations See SI page 2 for a listing of references for these data sources.	High quality offshore data is available globally. Most of these data at the global scale are the same ones commonly used at local scales. However: The time periods for analysis must be the same and are thus limited by the shortest data set (i.e., waves) Better local data cannot be incorporated (except for local SLR projections).	 • Often same as global data • Local observations in deep water (tidal gauges, waves, surge) • High resolution reanalysis (~1km) only by downscaling from regional or global data. • Local land subsidence estimates. • Local relative sea level rise projections similar to the ones for other scales 	Local observations can be of higher resolution. However: Usually there are no existing local observations. Even when there is better local data, the time series are too short for determining flood return periods required to estimate annual expected damages. The lack of local time series data principally explain why there currently are no local studies that calculate the annual expected flood reduction benefits of reefs.
2	Nearshore hydrodynamics	 • Propagation with Snell's law (accounts for directionality) • Bathymetry: global (~1km) • Global coral reef cover 	Downscaling technique required to propagate the full time series to the coral reefs location Comparatively low computational cost. However:	 • Dynamic propagation with numerical models (e.g. SWAN) • Bathymetry: high resolution (~10m) • High resolution coral 	All waves propagation processes are considered (linear and non-linear) Local observations available to calibrate the model High resolution outputs (waves and storm surge fields ~10m)

		map (~1km)	Non-linear propagation processes are not considered The quality of the bathymetry is not great (affects all regional flood models). No local observations are used to calibrate the model. The outputs are of relatively coarse resolution (~1km) relative to local studies.	reef cover (~10m)	However: Local observations may not be appropriate (limited location and time) for downscaling. Few shallow, tropical sites have the high resolution bathymetry needed to justify the use of these high resolution models. Field work is usually required data collection, which limits scale. High computational cost in mesh set-up and modelling.
3	Reef effect on flooding	 • Representative scale of transects: 2,000m • Reef model 1D: Energy balance equation model (breaking and friction) for waves and long wave equations linear model for storm surge • Breaking model includes a specific breaking index for coral reefs • Bathymetry: Global (1km) • Global Coral cover and roughness distribution 	Use of 1D phase averaged parametric models: Require only a simple, numerical mesh Reduce the computational costs to reasonable levels (to simulate years of waves and surge data). These models are consistent with the resolution of the bathymetric data available globally (~1km). However: 2-D wave propagation processes such as refraction-diffraction omitted Simple representation of ecosystems, only by its roughness: friction	 • 2D propagations • Reef effect is modelled directly using SWAN which solves a conservation of energy equation with a breaking model and friction equivalent to the global model • Reef effect can also be modelled with XBeach, which includes a similar breaking model and friction equivalent to the global model. • Bathymetry: Local (10m) • High resolution Coral cover and roughness 	All relevant processes are considered in wave propagation, including non-linear effects and infragravity waves Higher resolution ecosystem data can be used. However: Higher resolution bathymetric and ecosystem data are rarely available. Nonlinear effects and infragravity waves are based on a series of parameterizations that have been validated with only a limited number of field experiments. It is not computationally feasible yet to run these models at regional or global levels. Existing studies simulate a limited number of sea

			coefficient for coral reefs (Sheppard 2005).	distribution	states or storm events, because XBeach is computationally expensive to run for the full-time series necessary for a stochastic risk analysis. There is no major difference with the global approach in the way the effect of the coral reef is modelled in terms of breaking and friction. Breaking is parameterized using a breaking model and corals are represented using a friction coefficient with the actual bathymetry.
4	a) Estimate Flooding- Flood height at shoreline	 • The wave contribution to the total water level at the shoreline (Runup) is calculated using an empirical formulation • Reconstructions of historical time series of total water level over the full period • Extreme value analysis based on the time series reconstruction 	Runup formula considers the case of barred beaches which can be considered similar to coral reefs. The runup formula includes an infragravity wave component obtained empirically. Allows the reconstruction of the full-time series of the total water level required for stochastic risk analysis. Uncoupled calculation of flood height at shore and flooding inland provides future flexibility in re-running only flooding models (e.g., if new topographic or population data is developed).	 • The wave contribution to the total water level at the shoreline (Runup) is calculated using numerical models • Extreme value analysis of flood height time series is not necessary if coupling flood height calculation and flooding inland processes by using the same numerical model 	Coupled flood height and flooding calculation (“all in one”). Does include the contribution of infragravity waves using a process-based model. However: Applications only for extreme events or storms (high computational cost and limited number of events to be simulated). Quality of flood height estimates is strongly dependent on the quality of the bathymetric and topographic data.

		However: The calculation of the flood height at the shoreline and consequent flooding inland is obtained using different models (not coupled). This is a standard approach in global flood models.		
b) Estimate Flooding- inland	 • Modified bathtub approach with hydraulic connectivity • Topography: Global ~90m • Extreme values: Stationary Extreme value analysis 	Low computational cost. No mesh generation process required (same mesh as the topography model). Large scale domains can be run simultaneously (>1,000km). Does not require high resolution topographic data For coarse topographic data, it provides same results as site-based numerical flood models such as RFSM-EDA. It does not require very high-resolution field data (vegetation, buildings) However: Flood calculation does not consider terrain roughness, infiltration, rain, river flow.	 • Process-based models (RFSM-EDA) • Topography: high resolution (LIDAR 1m) 	Flood calculation considers terrain roughness, infiltration, rain, river flow. Non-stationary analysis (flood cover evolution in time). We can estimate flooding extent at any time step. However: High computational cost Complex mesh generation required. Only applicable to very small-scale domains simultaneously (<100km) It requires very high-resolution topography data (<5m) and field data (e.g., vegetation, buildings). No applications to date for calculating expected annual damages using the flooding height distribution. If waves and storm surge

			It may overestimate flooding. Stationary analysis (only initial and final flood scenarios). Downscaling errors due to differences between hazards resolution (~2km) and topographic resolutions (~90m).		propagation and flooding inland are simultaneously run, the quality of flooding is not comparable to process-based models for flooding such as RFSM or equivalent.
5	Assess Damages and Benefits	 • Global data on built capital (UNISDR) and people, resolution: ~5,000m • Representative vulnerability curve 	Global socioeconomic data freely available. Global data of population is improving in resolution (~100m). However: Built capital estimates at very low resolution (~5km). Difficult to find homogeneous global data of assets and people. Very general damage functions, not specific for each asset class or region. Downscaling errors if assets and people distribution grids are of different order of magnitude of the DEM data (<90m).	 • Census data, resolution: census unit or block level • Local Damage functions, for each type of building. 	Potential for higher accuracy local census data/ Downscaling errors avoided if assets and people distribution grids are of the order of magnitude of the topographic data (<5m) Very specific damage functions, classified for the different local assets, are possible. However: Better local socio-economic data (e.g., census) and damage curves are rarely available in particular for tropical developing countries. Heterogeneity and discrepancies between databases depending on local surveys processes.

With respect to the comments of Reviewer 3, the authors appear to have responded to most comments, but, really, Reviewer 3 should be asked whether they are happy with the response. I fully agree with Reviewer 3 that the conclusion ‘Overall the hotspot areas where reefs provide the most flood protection benefits are in storm belts with extensive, shallow and rugose coral reefs; land at low elevation; and assets concentrated on the coast’ is profoundly unsurprising.

We are glad that we could make a simple a clear statement about the global patterns. We can only reach such a point with extensive research and modeling. No other study could defend this point globally with models and analyses. We have further refined our statement about these results to make it clearer that we are making a statement about pattern but that our results go much further in indicating spatial variation around this pattern.

Line 127 “By integrating economic, ecological and hydrodynamic models, we show variation locally, nationally and regionally around the general pattern that reefs provide the most flood protection benefits in storm belts with extensive, shallow and rugose coral reefs; land at low elevation; and assets concentrated on the coast (Figure 5).”

I see the point made by Reviewer 1 that one can always criticise the global approach as being too simplistic and that there is a place for such approaches. But, is the Nature Group the correct outlet for such publications? Based on past evidence it clear is, as there are a plethora of such global-reach papers. Generally, they are not particularly rigorous, nor very insightful in terms of providing novel insights; however, they do make for good media soundbites and that is what seems to drive these publications: citability rather than scientific rigour and/or novelty. I believe this is an editorial decision, but in my view the quality of this paper does not meet the high standards I would expect of a Nature group publication.

The Nature Group publishes global studies regularly. We are sure that the Nature group will see the widespread impact of this paper. None of the reviews current or past points to any studies like ours. Indeed, we know that there is no similar study (from our extensive work with the ecosystem services program from the World Bank among others) and that it will get widespread use. We already get many requests from scientists, government agencies and businesses (e.g., multiple leading insurance companies) for access to the results and data, because of their uniqueness, rigor and clarity.